# Sequential Transfer in Multi-armed Bandit with Finite Set of Models

**Mohammad Gheshlaghi Azar** [*]
School of Computer Science
CMU

**Alessandro Lazaric** [†]
INRIA Lille - Nord Europe
Team SequeL

**Emma Brunskill** [*]
School of Computer Science
CMU

## Abstract

Learning from prior tasks and transferring that experience to improve future performance is critical for building lifelong learning agents. Although results in supervised and reinforcement learning show that transfer may significantly improve the learning performance, most of the literature on transfer is focused on batch learning tasks. In this paper we study the problem of *sequential transfer in online learning*, notably in the multi–armed bandit framework, where the objective is to minimize the total regret over a sequence of tasks by transferring knowledge from prior tasks. We introduce a novel bandit algorithm based on a method-of-moments approach for estimating the possible tasks and derive regret bounds for it.

## 1 Introduction

Learning from prior tasks and transferring that experience to improve future performance is a key aspect of intelligence, and is critical for building lifelong learning agents. Recently, multi-task and transfer learning received much attention in the supervised and reinforcement learning (RL) setting with both empirical and theoretical encouraging results (see recent surveys by Pan and Yang, 2010; Lazaric, 2011). Most of these works focused on scenarios where the tasks are batch learning problems, in which a training set is directly provided to the learner. On the other hand, the online learning setting (Cesa-Bianchi and Lugosi, 2006), where the learner is presented with samples in a sequential fashion, has been rarely considered (see Mann and Choe (2012); Taylor (2009) for examples in RL and Sec. E of Azar et al. (2013) for a discussion on related settings).

The multi–armed bandit (MAB) (Robbins, 1952) is a simple yet powerful framework formalizing the online learning with partial feedback problem, which encompasses a large number of applications, such as clinical trials, web advertisements and adaptive routing. In this paper we take a step towards understanding and providing formal bounds on transfer in stochastic MABs. We focus on a *sequential transfer* scenario where an (online) learner is acting in a series of tasks drawn from a stationary distribution over a finite set of MABs. The learning problem, within each task, can be seen as a standard MAB problem with a fixed number of steps. Prior to learning, the model parameters of each bandit problem are not known to the learner, nor does it know the distribution probability over the bandit problems. Also, we assume that the learner is not provided with the identity of the tasks throughout the learning. To act efficiently in this setting, it is crucial to define a mechanism for transferring knowledge across tasks. In fact, the learner may encounter the same bandit problem over and over throughout the learning, and an efficient algorithm should be able to leverage the knowledge obtained in previous tasks, when it is presented with the same problem again. To address this problem one can transfer the estimates of all the possible models from prior tasks to the current one. Once these models are accurately estimated, we show that an extension of the *UCB* algorithm (Auer et al., 2002) is able to efficiently exploit this prior knowledge and reduce the regret through tasks (Sec. 3).

---

[*]`{mazar,ebrun}@cs.cmu.edu`
[†]`alessandro.lazaric@inria.fr`

The main contributions of this paper are two-fold: **(i)** we introduce the *tUCB* algorithm which transfers the model estimates across the tasks and uses this knowledge to achieve a better performance than *UCB*. We also prove that the new algorithm is guaranteed to perform as well as *UCB* in early episodes, thus avoiding any *negative transfer* effect, and then to approach the performance of the ideal case when the models are all known in advance (Sec. 4.4). **(ii)** To estimate the models we rely on a new variant of method of moments, robust tensor power method (RTP) (Anandkumar et al., 2013, 2012b) and extend it to the multi-task bandit setting[1]:we prove that *RTP* provides a consistent estimate of the means of all arms (for all models) as long as they are pulled at least three times per task and prove sample complexity bounds for it (Sec. 4.2). Finally, we report some preliminary results on synthetic data confirming the theoretical findings (Sec. 5). An extended version of this paper containing proofs and additional comments is available in (Azar et al., 2013).

## 2   Preliminaries

We consider a stochastic MAB problem defined by a set of arms $\mathcal{A} = \{1, \ldots, K\}$, where each arm $i \in \mathcal{A}$ is characterized by a distribution $\nu_i$ and the samples (rewards) observed from each arm are independent and identically distributed. We focus on the setting where there exists a set of models $\Theta = \{\theta = (\nu_1, \ldots, \nu_K)\}, |\Theta| = m$, which contains all the possible bandit problems. We denote the mean of an arm $i$, the best arm, and the best value of a model $\theta \in \Theta$ respectively by $\mu_i(\theta)$, $i_*(\theta)$, $\mu_*(\theta)$. We define the arm gap of an arm $i$ for a model $\theta$ as $\Delta_i(\theta) = \mu_*(\theta) - \mu_i(\theta)$, while the model gap for an arm $i$ between two models $\theta$ and $\theta'$ is defined as $\Gamma_i(\theta, \theta') = |\mu_i(\theta) - \mu_i(\theta')|$. We also assume that arm rewards are bounded in $[0, 1]$. We consider the sequential transfer setting where at each episode $j$ the learner interacts with a task $\bar{\theta}^j$, drawn from a distribution $\rho$ over $\Theta$, for $n$ steps. The objective is to minimize the (pseudo-)regret $\mathcal{R}_J$ over $J$ episodes measured as the difference between the rewards obtained by pulling $i_*(\bar{\theta}^j)$ and those achieved by the learner:

$$\mathcal{R}_J = \sum_{j=1}^{J} \mathcal{R}_n^j = \sum_{j=1}^{J} \sum_{i \neq i^*} T_{i,n}^j \Delta_i(\bar{\theta}^j), \tag{1}$$

where $T_{i,n}^j$ is the number of pulls to arm $i$ after $n$ steps of episode $j$. We also introduce some tensor notation. Let $X \in \mathbb{R}^K$ be a random realization of the rewards of all arms from a random model. All the realizations are i.i.d. conditional on a model $\bar{\theta}$ and $\mathbb{E}[X|\theta = \bar{\theta}] = \mu(\theta)$, where the $i$-th component of $\mu(\theta) \in \mathbb{R}^K$ is $[\mu(\theta)]_i = \mu_i(\theta)$. Given realizations $X^1$, $X^2$ and $X^3$, we define the second moment matrix $M_2 = \mathbb{E}[X^1 \otimes X^2]$ such that $[M_2]_{i,j} = \mathbb{E}[X_i^1 X_j^2]$ and the third moment tensor $M_3 = \mathbb{E}[X^1 \otimes X^2 \otimes X^3]$ such that $[M_2]_{i,j,l} = \mathbb{E}[X_i^1 X_j^2 X_l^3]$. Since the realizations are conditionally independent, we have that, for every $\theta \in \Theta$, $\mathbb{E}[X^1 \otimes X^2|\theta] = \mathbb{E}[X^1|\theta] \otimes \mathbb{E}[X^2|\theta] = \mu(\theta) \otimes \mu(\theta)$ and this allows us to rewrite the second and third moments as $M_2 = \sum_\theta \rho(\theta)\mu(\theta)^{\otimes 2}, M_3 = \sum_\theta \rho(\theta)\mu(\theta)^{\otimes 3}$, where $v^{\otimes p} = v \otimes v \otimes \cdots v$ is the $p$-th tensor power. Let $A$ be a $3^{\text{rd}}$ order member of the tensor product of the Euclidean space $\mathbb{R}^K$ (as $M_3$), then we define the multilinear map as follows. For a set of three matrices $\{V_i \in \mathbb{R}^{K \times m}\}_{1 \leq i \leq 3}$, the $(i_1, i_2, i_3)$ entry in the 3-way array representation of $A(V_1, V_2, V_3) \in \mathbb{R}^{m \times m \times m}$ is $[A(V_1, V_2, V_3)]_{i_1, i_2, i_3} := \sum_{1 \leq j_1, j_2, j_3 \leq n} A_{j_1, j_2, j_3} [V_1]_{j_1, i_1} [V_2]_{j_2, i_2} [V_3]_{j_3, i_3}$. We also use different norms: the Euclidean norm $\|\cdot\|$; the Frobenius norm $\|\cdot\|_F$; the matrix max-norm $\|A\|_{\max} = \max_{ij} |[A]_{ij}|$.

## 3   Multi-arm Bandit with Finite Models

Before considering the transfer problem, we show that a simple variation to *UCB* allows us to effectively exploit the knowledge of $\Theta$ and obtain a significant reduction in the regret. The *mUCB* (model-UCB) algorithm in Fig. 1 takes as input a set of models $\Theta$ including the current (unknown) model $\bar{\theta}$. At each step $t$, the algorithm computes a subset $\Theta_t \subseteq \Theta$ containing only the models whose means $\mu_i(\theta)$ are *compatible* with the current estimates $\hat{\mu}_{i,t}$ of the means $\mu_i(\bar{\theta})$ of the current model, obtained averaging

> **Require:** Set of models $\Theta$, number of steps $n$
>   **for** $t = 1, \ldots, n$ **do**
>     Build $\Theta_t = \{\theta : \forall i, |\mu_i(\theta) - \hat{\mu}_{i,t}| \leq \varepsilon_{i,t}\}$
>     Select $\theta_t = \arg \max_{\theta \in \Theta_t} \mu_*(\theta)$
>     Pull arm $I_t = i_*(\theta_t)$
>     Observe sample $x_{I_t}$ and update
>   **end for**

Figure 1: The *mUCB* algorithm.

[1]Notice that estimating the models involves solving a latent variable model estimation problem, for which RTP is the state-of-the-art.

$T_{i,t}$ pulls, and their uncertainty $\varepsilon_{i,t}$ (see Eq. 2 for an explicit definition of this term). Notice that it is enough that one arm does not satisfy the compatibility condition to discard a model $\theta$. Among all the models in $\Theta_t$, *mUCB* first selects the model with the largest optimal value and then it pulls its corresponding optimal arm. This choice is coherent with the *optimism in the face of uncertainty* principle used in UCB-based algorithms, since *mUCB* always pulls the optimal arm corresponding to the optimistic model compatible with the current estimates $\hat{\mu}_{i,t}$. We show that *mUCB* incurs a regret which is never worse than *UCB* and it is often significantly smaller.

We denote the set of arms which are optimal for at least a model in a set $\Theta'$ as $\mathcal{A}_*(\Theta') = \{i \in \mathcal{A} : \exists \theta \in \Theta' : i_*(\theta) = i\}$. The set of models for which the arms in $\mathcal{A}'$ are optimal is $\Theta(\mathcal{A}') = \{\theta \in \Theta : \exists i \in \mathcal{A}' : i_*(\theta) = i\}$. The set of optimistic models for a given model $\bar{\theta}$ is $\Theta_+ = \{\theta \in \Theta : \mu_*(\theta) \geq \mu_*(\bar{\theta})\}$, and their corresponding optimal arms $\mathcal{A}_+ = \mathcal{A}_*(\Theta_+)$. The following theorem bounds the expected regret (similar bounds hold in high probability). The lemmas and proofs (using standard tools from the bandit literature) are available in Sec. B of Azar et al. (2013).

**Theorem 1.** *If mUCB is run with $\delta = 1/n$, a set of $m$ models $\Theta$ such that the $\bar{\theta} \in \Theta$ and*

$$\varepsilon_{i,t} = \sqrt{\log(mn^2/\delta)/(2T_{i,t-1})}, \tag{2}$$

*where $T_{i,t-1}$ is the number of pulls to arm $i$ at the beginning of step $t$, then its expected regret is*

$$\mathbb{E}[\mathcal{R}_n] \leq K + \sum\nolimits_{i \in \mathcal{A}_+} \frac{2\Delta_i(\bar{\theta}) \log(mn^3)}{\min_{\theta \in \Theta_{+,i}} \Gamma_i(\theta, \bar{\theta})^2} \leq K + \sum\nolimits_{i \in \mathcal{A}_+} \frac{2\log(mn^3)}{\min_{\theta \in \Theta_{+,i}} \Gamma_i(\theta, \bar{\theta})}, \tag{3}$$

*where $\mathcal{A}_+ = \mathcal{A}_*(\Theta_+)$ is the set of arms which are optimal for at least one optimistic model $\Theta_+$ and $\Theta_{+,i} = \{\theta \in \Theta_+ : i_*(\theta) = i\}$ is the set of optimistic models for which $i$ is the optimal arm.*

**Remark (comparison to *UCB*).** The *UCB* algorithm incurs a regret

$$\mathbb{E}[\mathcal{R}_n(\text{UCB})] \leq O\Big(\sum\nolimits_{i \in \mathcal{A}} \frac{\log n}{\Delta_i(\bar{\theta})}\Big) \leq O\Big(K \frac{\log n}{\min_i \Delta_i(\bar{\theta})}\Big).$$

We see that *mUCB* displays two major improvements. The regret in Eq. 3 can be written as

$$\mathbb{E}[\mathcal{R}_n(\text{mUCB})] \leq O\Big(\sum\nolimits_{i \in \mathcal{A}_+} \frac{\log n}{\min_{\theta \in \Theta_{+,i}} \Gamma_i(\theta, \bar{\theta})}\Big) \leq O\Big(|\mathcal{A}_+| \frac{\log n}{\min_i \min_{\theta \in \Theta_{+,i}} \Gamma_i(\theta, \bar{\theta})}\Big).$$

This result suggests that *mUCB* tends to discard all the models in $\Theta_+$ from the most optimistic down to the actual model $\bar{\theta}$ which, with high-probability, is never discarded. As a result, even if other models are still in $\Theta_t$, the optimal arm of $\bar{\theta}$ is pulled until the end. This significantly reduces the set of arms which are actually pulled by *mUCB* and the previous bound only depends on the number of arms in $\mathcal{A}_+$, which is $|\mathcal{A}_+| \leq |\mathcal{A}_*(\Theta)| \leq K$. Furthermore, for all arms $i$, the minimum gap $\min_{\theta \in \Theta_{+,i}} \Gamma_i(\theta, \bar{\theta})$ is guaranteed to be larger than the arm gap $\Delta_i(\bar{\theta})$ (see Lem. 4 in Sec. B of Azar et al. (2013)), thus further improving the performance of *mUCB* w.r.t. *UCB*.

## 4 Online Transfer with Unknown Models

We now consider the case when the set of models is unknown and the regret is cumulated over multiple tasks drawn from $\rho$ (Eq. 1). We introduce *tUCB* (transfer-UCB) which transfers estimates of $\Theta$, whose accuracy is improved through episodes using a method-of-moments approach.

### 4.1 The transfer-UCB Bandit Algorithm

Fig. 2 outlines the structure of our online transfer bandit algorithm *tUCB* (transfer-UCB). The algorithm uses two sub-algorithms, the bandit algorithm *umUCB* (*uncertain model-UCB*), whose objective is to minimize the regret at each episode, and *RTP* (*robust tensor power* method) which at each episode $j$ computes an estimate $\{\hat{\mu}_i^j(\theta)\}$ of the arm means of all the models. The bandit algorithm *umUCB* in Fig. 3 is an extension of the *mUCB* algorithm. It first computes a set of models $\Theta_t^j$ whose means $\hat{\mu}_i(\theta)$ are compatible with the current estimates $\hat{\mu}_{i,t}$. However, unlike the case where the exact models are available, here the models themselves are estimated and the uncertainty $\varepsilon^j$ in their means (provided as input to *umUCB*) is taken into account in the definition of $\Theta_t^j$. Once

<div>

**Require:** number of arms $K$, number of
models $m$, constant $C(\theta)$.
**Initialize** estimated models $\Theta^1 = \{\hat{\mu}_i^1(\theta)\}_{i,\theta}$, samples $R \in \mathbb{R}^{J \times K \times n}$
**for** $j = 1, 2, \ldots, J$ **do**
    Run $R^j = \text{umUCB}(\Theta^j, n)$
    Run $\Theta^{j+1} = \text{RTP}(R, m, K, j, \delta)$
**end for**

Figure 2: The *tUCB* algorithm.

**Require:** set of models $\Theta^j$, num. steps $n$
Pull each arm three times
**for** $t = 3K + 1, \ldots, n$ **do**
    Build $\Theta_t^j = \{\theta : \forall i, |\hat{\mu}_i^j(\theta) - \hat{\mu}_{i,t}| \leq \varepsilon_{i,t} + \varepsilon^j\}$
    Compute $B_t^j(i; \theta) = \min\{(\hat{\mu}_i^j(\theta) + \varepsilon^j), (\hat{\mu}_{i,t} + \varepsilon_{i,t})\}$
    Compute $\theta_t^j = \arg\max_{\theta \in \Theta_t^j} \max_i B_t^j(i; \theta)$
    Pull arm $I_t = \arg\max_i B_t^j(i; \theta_t^j)$
    Observe sample $R(I_t, T_{i,t}) = x_{I_t}$ and update
**end for**
**return** Samples $R$

Figure 3: The *umUCB* algorithm.

**Require:** samples $R \in \mathbb{R}^{j \times n}$, number of models $m$ and arms $K$, episode $j$
Estimate the second and third moment $\widehat{M_2}$ and $\widehat{M_3}$ using the reward samples from $R$ (Eq. 4)
Compute $\widehat{D} \in \mathbb{R}^{m \times m}$ and $\widehat{U} \in \mathbb{R}^{K \times m}$ ($m$ largest eigenvalues and eigenvectors of $\widehat{M_2}$ resp.)
Compute the whitening mapping $\widehat{W} = \widehat{U}\widehat{D}^{-1/2}$ and the tensor $\widehat{T} = \widehat{M_3}(\widehat{W}, \widehat{W}, \widehat{W})$
Plug $\widehat{T}$ in Alg. 1 of Anandkumar et al. (2012b) and compute eigen-vectors/values $\{\widehat{v}(\theta)\}, \{\widehat{\lambda}(\theta)\}$
Compute $\widehat{\mu}^j(\theta) = \widehat{\lambda}(\theta)(\widehat{W}^\mathsf{T})^+\widehat{v}(\theta)$ for all $\theta \in \Theta$
**return** $\Theta^{j+1} = \{\widehat{\mu}^j(\theta) : \theta \in \Theta\}$

Figure 4: The robust tensor power (*RTP*) method (Anandkumar et al., 2012b).

</div>

the active set is computed, the algorithm computes an upper-confidence bound on the value of each
arm $i$ for each model $\theta$ and returns the best arm for the most optimistic model. Unlike in *mUCB*,
due to the uncertainty over the model estimates, a model $\theta$ might have more than one optimal arm,
and an upper-confidence bound on the mean of the arms $\hat{\mu}_i(\theta) + \varepsilon^j$ is used together with the upper-
confidence bound $\hat{\mu}_{i,t} + \varepsilon_{i,t}$, which is directly derived from the samples observed so far from arm
$i$. This guarantees that the $B$-values are always consistent with the samples generated from the ac-
tual model $\bar{\theta}^j$. Once *umUCB* terminates, *RTP* (Fig. 4) updates the estimates of the model means
$\widehat{\mu}^j(\theta) = \{\hat{\mu}_i^j(\theta)\}_i \in \mathbb{R}^K$ using the samples obtained from each arm $i$. At the beginning of each task
*umUCB* pulls all the arms 3 times, since *RTP* needs at least 3 samples from each arm to accurately
estimate the 2$^{\text{nd}}$ and 3$^{\text{rd}}$ moments (Anandkumar et al., 2012b). More precisely, *RTP* uses all the
reward samples generated up to episode $j$ to estimate the 2$^{\text{nd}}$ and 3$^{\text{rd}}$ moments (see Sec. 2) as

$$\widehat{M_2} = j^{-1}\sum\nolimits_{l=1}^{j} \overline{\mu}_{1l} \otimes \overline{\mu}_{2l}, \qquad \text{and} \qquad \widehat{M_3} = j^{-1}\sum\nolimits_{l=1}^{j} \overline{\mu}_{1l} \otimes \overline{\mu}_{2l} \otimes \overline{\mu}_{3l}, \qquad (4)$$

where the vectors $\overline{\mu}_{1l}, \overline{\mu}_{2l}, \overline{\mu}_{3l} \in \mathbb{R}^K$ are obtained by dividing the $T_{i,n}^l$ samples observed from arm
$i$ in episode $l$ in three batches and taking their average (e.g., $[\overline{\mu}_{1l}]_i$ is the average of the first $T_{i,n}^l/3$
samples).[2] Since $\overline{\mu}_{1l}, \overline{\mu}_{2l}, \overline{\mu}_{3l}$ are independent estimates of $\mu(\bar{\theta}^l)$, $\widehat{M_2}$ and $\widehat{M_3}$ are consistent esti-
mates of the second and third moments $M_2$ and $M_3$. *RTP* relies on the fact that the model means
$\mu(\theta)$ can be recovered from the spectral decomposition of the symmetric tensor $T = M_3(W, W, W)$,
where $W$ is a whitening matrix for $M_2$, i.e., $M_2(W, W) = \mathbf{I}^{m \times m}$ (see Sec. 2 for the defini-
tion of the mapping $A(V_1, V_2, V_3)$). Anandkumar et al. (2012b) (Thm. 4.3) have shown that un-
der some mild assumption (see later Assumption 1) the model means $\{\mu(\theta)\}$, can be obtained as
$\mu(\theta) = \lambda(\theta)Bv(\theta)$, where $(\lambda(\theta), v(\theta))$ is a pair of eigenvector/eigenvalue for the tensor $T$ and
$B := (W^\mathsf{T})^+$. Thus the *RTP* algorithm estimates the eigenvectors $\widehat{v}(\theta)$ and the eigenvalues $\widehat{\lambda}(\theta)$, of
the $m \times m \times m$ tensor $\widehat{T} := \widehat{M_3}(\widehat{W}, \widehat{W}, \widehat{W})$.[3] Once $\widehat{v}(\theta)$ and $\widehat{\lambda}(\theta)$ are computed, the estimated
mean vector $\widehat{\mu}^j(\theta)$ is obtained by the inverse transformation $\widehat{\mu}^j(\theta) = \widehat{\lambda}(\theta)\widehat{B}\widehat{v}(\theta)$, where $\widehat{B}$ is the
pseudo inverse of $\widehat{W}^\mathsf{T}$ (for a detailed description of RTP algorithm see Anandkumar et al., 2012b).

general $\widehat{W}$ is not unique. Here, we choose $\widehat{W} = \widehat{U}\widehat{D}^{-1/2}$, where $\widehat{D} \in \mathbb{R}^{m \times m}$ is a diagonal matrix consisting
of the $m$ largest eigenvalues of $\widehat{M_2}$ and $\widehat{U} \in \mathbb{R}^{K \times m}$ has the corresponding eigenvectors as its columns.

## 4.2 Sample Complexity of the Robust Tensor Power Method

*umUCB* requires as input $\varepsilon^j$, i.e., the uncertainty of the model estimates. Therefore we need sample complexity bounds on the accuracy of $\{\hat{\mu}_i(\theta)\}$ computed by *RTP*. The performance of *RTP* is directly affected by the error of the estimates $\widehat{M}_2$ and $\widehat{M}_3$ w.r.t. the true moments. In Thm. 2 we prove that, as the number of tasks $j$ grows, this error rapidly decreases with the rate of $\sqrt{1/j}$. This result provides us with an upper-bound on the error $\varepsilon^j$ needed for building the confidence intervals in *umUCB*. The following definition and assumption are required for our result.

**Definition 1.** *Let* $\Sigma_{M_2} = \{\sigma_1, \sigma_2, \ldots, \sigma_m\}$ *be the set of* $m$ *largest eigenvalues of the matrix* $M_2$. *Define* $\sigma_{\min} := \min_{\sigma \in \Sigma_{M_2}} \sigma$, $\sigma_{\max} := \max_{\sigma \in \Sigma_{M_2}} \sigma$ *and* $\lambda_{\max} := \max_\theta \lambda(\theta)$. *Define the minimum gap between the distinct eigenvalues of* $M_2$ *as* $\Gamma_\sigma := \min_{\sigma_i \neq \sigma_l}(|\sigma_i - \sigma_l|)$.

**Assumption 1.** *The mean vectors* $\{\mu(\theta)\}_\theta$ *are linear independent and* $\rho(\theta) > 0$ *for all* $\theta \in \Theta$.

We now state our main result which is in the form of a high probability bound on the estimation error of mean reward vector of every model $\theta \in \Theta$.

**Theorem 2.** *Pick* $\delta \in (0, 1)$. *Let* $C(\Theta) := C_3 \lambda_{\max} \sqrt{\sigma_{\max}/\sigma_{\min}^3} (\sigma_{\max}/\Gamma_\sigma + 1/\sigma_{\min} + 1/\sigma_{\max})$, *where* $C_3 > 0$ *is a universal constant. Then under Assumption 1 there exist constants* $C_4 > 0$ *and a permutation* $\pi$ *on* $\Theta$, *such that for all* $\theta \in \Theta$, *we have w.p.* $1 - \delta$

$$\|\mu(\theta) - \hat{\mu}^j(\pi(\theta))\| \leq \varepsilon_j \triangleq C(\Theta) K^{2.5} m^2 \sqrt{\frac{\log(K/\delta)}{j}} \quad after \quad j \geq \frac{C_4 m^5 K^6 \log(K/\delta)}{\min(\sigma_{\min}, \Gamma_\sigma)^2 \sigma_{\min}^3 \lambda_{\min}^2}. \quad (5)$$

**Remark (computation of $C(\Theta)$).** As illustrated in Fig. 3, *umUCB* relies on the estimates $\hat{\mu}^j(\theta)$ and on their accuracy $\varepsilon^j$. Although the bound reported in Thm. 2 provides an upper confidence bound on the error of the estimates, it contains terms which are not computable in general (e.g., $\sigma_{\min}$). In practice, $C(\Theta)$ should be considered as a parameter of the algorithm. This is not dissimilar from the parameter usually introduced in the definition of $\varepsilon_{i,t}$ in front of the square-root term in *UCB*.

## 4.3 Regret Analysis of *umUCB*

We now analyze the regret of *umUCB* when an estimated set of models $\Theta^j$ is provided as input. At episode $j$, for each model $\theta$ we define the set of non-dominated arms (i.e., potentially optimal arms) as $\mathcal{A}_*^j(\theta) = \{i \in \mathcal{A} : \nexists i', \hat{\mu}_i^j(\theta) + \varepsilon^j < \hat{\mu}_{i'}^j(\theta) - \varepsilon^j\}$. Among the non-dominated arms, when the actual model is $\bar{\theta}^j$, the set of optimistic arms is $\mathcal{A}_+^j(\theta; \bar{\theta}^j) = \{i \in \mathcal{A}_*^j(\theta) : \hat{\mu}_i^j(\theta) + \varepsilon^j \geq \mu^*(\bar{\theta}^j)\}$. As a result, the set of optimistic models is $\Theta_+^j(\bar{\theta}^j) = \{\theta \in \Theta : \mathcal{A}_+^j(\theta; \bar{\theta}^j) \neq \emptyset\}$. In some cases, because of the uncertainty in the model estimates, unlike in *mUCB*, not all the models $\theta \neq \bar{\theta}^j$ can be discarded, not even at the end of a very long episode. Among the optimistic models, the set of models that cannot be discarded is defined as $\widetilde{\Theta}_+^j(\bar{\theta}^j) = \{\theta \in \Theta_+^j(\bar{\theta}^j) : \forall i \in \mathcal{A}_+^j(\theta; \bar{\theta}^j), |\hat{\mu}_i^j(\theta) - \mu_i(\bar{\theta}^j)| \leq \varepsilon^j\}$. Finally, when we want to apply the previous definitions to a set of models $\Theta'$ instead of single model we have, e.g., $\mathcal{A}_*^j(\Theta'; \bar{\theta}^j) = \bigcup_{\theta \in \Theta'} \mathcal{A}_*^j(\theta; \bar{\theta}^j)$.

The proof of the following results are available in Sec. D of Azar et al. (2013), here we only report the number of pulls, and the corresponding regret bound.

**Corollary 1.** *If at episode $j$ umUCB is run with $\varepsilon_{i,t}$ as in Eq. 2 and $\varepsilon^j$ as in Eq. 5 with a parameter $\delta' = \delta/2K$, then for any arm $i \in \mathcal{A}$, $i \neq i_*(\bar{\theta}^j)$ is pulled $T_{i,n}$ times such that*

$$\begin{cases} T_{i,n} \leq \min\left\{\frac{2\log\left(2mKn^2/\delta\right)}{\Delta_i(\bar{\theta}^j)^2}, \frac{\log\left(2mKn^2/\delta\right)}{2\min_{\theta \in \Theta_{i,+}^j(\bar{\theta}^j)} \widehat{\Gamma}_i(\theta; \bar{\theta}^j)^2}\right\} + 1 & if\ i \in \mathcal{A}_1^j \\ T_{i,n} \leq 2\log\left(2mKn^2/\delta\right)/(\Delta_i(\bar{\theta}^j)^2) + 1 & if\ i \in \mathcal{A}_2^j \\ T_{i,n} = 0 & otherwise \end{cases}$$

*w.p.* $1 - \delta$, *where* $\Theta_{i,+}^j(\bar{\theta}^j) = \{\theta \in \Theta_+^j(\bar{\theta}^j) : i \in \mathcal{A}_+(\theta; \bar{\theta}^j)\}$ *is the set of models for which $i$ is among their optimistic non-dominated arms,* $\widehat{\Gamma}_i(\theta; \bar{\theta}^j) = \Gamma_i(\theta, \bar{\theta}^j)/2 - \varepsilon^j$, $\mathcal{A}_1^j = \mathcal{A}_+^j(\Theta_+^j(\bar{\theta}^j); \bar{\theta}^j) - \mathcal{A}_+^j(\widetilde{\Theta}_+^j(\bar{\theta}^j); \bar{\theta}^j)$ *(i.e., set of arms only proposed by models that can be discarded), and* $\mathcal{A}_2^j = \mathcal{A}_+^j(\widetilde{\Theta}_+^j(\bar{\theta}^j); \bar{\theta}^j)$ *(i.e., set of arms only proposed by models that cannot be discarded).*

The previous corollary states that arms which cannot be optimal for any optimistic model (i.e., the optimistic non-dominated arms) are never pulled by *umUCB*, which focuses only on arms in $i \in \mathcal{A}_+^j(\Theta_+^j(\bar{\theta}^j); \bar{\theta}^j)$. Among these arms, those that may help to remove a model from the active set (i.e., $i \in \mathcal{A}_1^j$) are potentially pulled less than *UCB*, while the remaining arms, which are optimal for the models that cannot be discarded (i.e., $i \in \mathcal{A}_2^j$), are simply pulled according to a *UCB* strategy. Similar to *mUCB*, *umUCB* first pulls the arms that are more *optimistic* until either the active set $\Theta_t^j$ changes or they are no longer optimistic (because of the evidence from the actual samples). We are now ready to derive the per-episode regret of *umUCB*.

**Theorem 3.** *If umUCB is run for $n$ steps on the set of models $\Theta^j$ estimated by RTP after $j$ episodes with $\delta = 1/n$, and the actual model is $\bar{\theta}^j$, then its expected regret (w.r.t. the random realization in episode $j$ and conditional on $\bar{\theta}^j$) is*

$$\mathbb{E}[\mathcal{R}_n^j] \leq K + \sum_{i \in \mathcal{A}_1^j} \log\left(2mKn^3\right) \min\left\{2/\Delta_i(\bar{\theta}^j)^2, 1/\left(2\min_{\theta \in \Theta_{i,+}^j(\bar{\theta}^j)} \widehat{\Gamma}_i(\theta; \bar{\theta}^j)^2\right)\right\} \Delta_i(\bar{\theta}^j)$$
$$+ \sum_{i \in \mathcal{A}_2^j} 2\log\left(2mKn^3\right)/\Delta_i(\bar{\theta}^j).$$

**Remark (negative transfer).** The transfer of knowledge introduces a bias in the learning process which is often beneficial. Nonetheless, in many cases transfer may result in a bias towards wrong solutions and a worse learning performance, a phenomenon often referred to as *negative transfer*. The first interesting aspect of the previous theorem is that *umUCB* is guaranteed to never perform worse than *UCB* itself. This implies that *tUCB* never suffers from negative transfer, even when the set $\Theta^j$ contains highly uncertain models and might bias *umUCB* to pull suboptimal arms.

**Remark (improvement over *UCB*).** In Sec. 3 we showed that *mUCB* exploits the knowledge of $\Theta$ to focus on a restricted set of arms which are pulled less than *UCB*. In *umUCB* this improvement is not as clear, since the models in $\Theta$ are not known but are estimated online through episodes. Yet, similar to *mUCB*, *umUCB* has the two main sources of potential improvement w.r.t. to *UCB*. As illustrated by the regret bound in Thm. 3, *umUCB* focuses on arms in $\mathcal{A}_1^j \cup \mathcal{A}_2^j$ which is potentially a smaller set than $\mathcal{A}$. Furthermore, the number of pulls to arms in $\mathcal{A}_1^j$ is smaller than for *UCB* whenever the estimated model gap $\widehat{\Gamma}_i(\theta; \bar{\theta}^j)$ is bigger than $\Delta_i(\bar{\theta}^j)$. Eventually, *umUCB* reaches the same performance (and improvement over *UCB*) as *mUCB* when $j$ is big enough. In fact, the set of optimistic models reduces to the one used in *mUCB* (i.e., $\Theta_+^j(\bar{\theta}^j) \equiv \Theta_+(\bar{\theta}^j)$) and all the optimistic models have only optimal arms (i.e., for any $\theta \in \Theta_+$ the set of non-dominated optimistic arms is $\mathcal{A}_+(\theta; \bar{\theta}^j) = \{i_*(\theta)\}$), which corresponds to $\mathcal{A}_1^j \equiv \mathcal{A}_*(\Theta_+(\bar{\theta}^j))$ and $\mathcal{A}_2^j \equiv \{i_*(\bar{\theta}^j)\}$, which matches the condition of *mUCB*. For instance, for any model $\theta$, in order to have $\mathcal{A}_*(\theta) = \{i_*(\theta)\}$, for any arm $i \neq i_*(\theta)$ we need that $\hat{\mu}_i^j(\theta) + \varepsilon^j \leq \hat{\mu}_{i_*(\theta)}^j(\theta) - \varepsilon^j$. Thus after

$$j \geq \frac{2C(\Theta)}{\min_{\bar{\theta} \in \Theta} \min_{\theta \in \Theta_+(\bar{\theta})} \min_i \Delta_i(\theta)^2} + 1.$$

episodes, all the optimistic models have only one optimal arm independently from the actual identity of the model $\bar{\theta}^j$. Although this condition may seem restrictive, in practice *umUCB* starts improving over *UCB* much earlier, as illustrated in the numerical simulation in Sec. 5.

### 4.4 Regret Analysis of *tUCB*

Given the previous results, we derive the bound on the cumulative regret over $J$ episodes (Eq. 1).

**Theorem 4.** *If tUCB is run over $J$ episodes of $n$ steps in which the tasks $\bar{\theta}^j$ are drawn from a fixed distribution $\rho$ over a set of models $\Theta$, then its cumulative regret is*

$$\mathcal{R}_J \leq JK + \sum_{j=1}^J \sum_{i \in \mathcal{A}_1^j} \min\left\{\frac{2\log\left(2mKn^2/\delta\right)}{\Delta_i(\bar{\theta}^j)^2}, \frac{\log\left(2mKn^2/\delta\right)}{2\min_{\theta \in \Theta_{i,+}^j(\bar{\theta}^j)} \widehat{\Gamma}_i^j(\theta; \bar{\theta}^j)^2}\right\} \Delta_i(\bar{\theta}^j)$$

$$+ \sum_{j=1}^J \sum_{i \in \mathcal{A}_2^j} \frac{2\log\left(2mKn^2/\delta\right)}{\Delta_i(\bar{\theta}^j)},$$

*w.p. $1 - \delta$ w.r.t. the randomization over tasks and the realizations of the arms in each episode.*

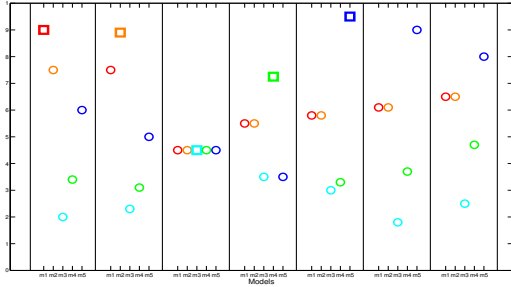

Figure 5: Set of models $\Theta$.

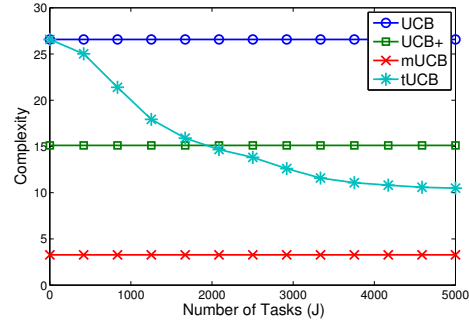

Figure 6: Complexity over tasks.

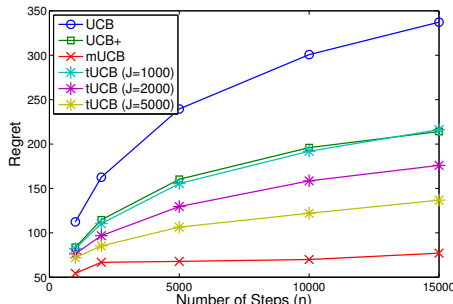

Figure 7: Regret of *UCB*, *UCB+*, *mUCB*, and *tUCB* (avg. over episodes) vs episode length.

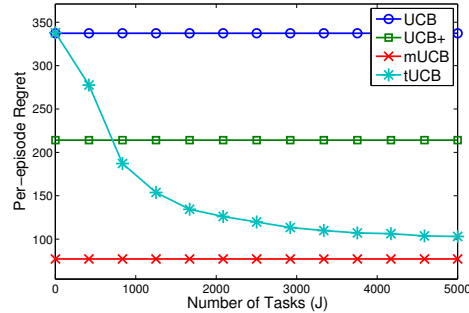

Figure 8: Per-episode regret of *tUCB*.

This result immediately follows from Thm. 3 and it shows a linear dependency on the number of episodes $J$. This dependency is the price to pay for not knowing the identity of the current task $\bar{\theta}^j$. If the task was revealed at the beginning of the task, a bandit algorithm could simply cluster all the samples coming from the same task and incur a much smaller cumulative regret with a logarithmic dependency on episodes and steps, i.e., $\log(nJ)$. Nonetheless, as discussed in the previous section, the cumulative regret of *tUCB* is never worse than for *UCB* and as the number of tasks increases it approaches the performance of *mUCB*, which fully exploits the prior knowledge of $\Theta$.

## 5  Numerical Simulations

In this section we report preliminary results of *tUCB* on synthetic data. The objective is to illustrate and support the previous theoretical findings. We define a set $\Theta$ of $m = 5$ MAB problems with $K = 7$ arms each, whose means $\{\mu_i(\theta)\}_{i,\theta}$ are reported in Fig. 5 (see Sect. F in Azar et al. (2013) for the actual values), where each model has a different color and squares correspond to optimal arms (e.g., arm 2 is optimal for model $\theta_2$). This set of models is chosen to be challenging and illustrate some interesting cases useful to understand the functioning of the algorithm.[4] Models $\theta_1$ and $\theta_2$ only differ in their optimal arms and this makes it difficult to distinguish them. For arm 3 (which is optimal for model $\theta_3$ and thus potentially selected by *mUCB*), all the models share exactly the same mean value. This implies that no model can be discarded by pulling it. Although this might suggest that *mUCB* gets stuck in pulling arm 3, we showed in Thm. 1 that this is not the case. Models $\theta_1$ and $\theta_5$ are challenging for *UCB* since they have small minimum gap. Only 5 out of the 7 arms are actually optimal for a model in $\Theta$. Thus, we also report the performance of *UCB+* which, under the assumption that $\Theta$ is known, immediately discards all the arms which are not optimal ($i \notin \mathcal{A}^*$) and performs *UCB* on the remaining arms. The model distribution is uniform, i.e., $\rho(\theta) = 1/m$.

Before discussing the transfer results, we compare *UCB*, *UCB+*, and *mUCB*, to illustrate the advantage of the prior knowledge of $\Theta$ w.r.t. *UCB*. Fig. 7 reports the per-episode regret of the three

algorithms for episodes of different length $n$ (the performance of *tUCB* is discussed later). The results are averaged over all the models in $\Theta$ and over 200 runs each. All the algorithms use the same confidence bound $\varepsilon_{i,t}$. The performance of *mUCB* is significantly better than both *UCB*, and *UCB+*, thus showing that *mUCB* makes an efficient use of the prior of knowledge of $\Theta$. Furthermore, in Fig. 6 the horizontal lines correspond to the value of the regret bounds up to the $n$ dependent terms and constants[5] for the different models in $\Theta$ averaged w.r.t. $\rho$ for the three algorithms (the actual values for the different models are in the supplementary material). These values show that the improvement observed in practice is accurately predicated by the upper-bounds derived in Thm. 1.

We now move to analyze the performance of *tUCB*. In Fig. 8 we show how the per-episode regret changes through episodes for a transfer problem with $J = 5000$ tasks of length $n = 5000$. In *tUCB* we used $\varepsilon^j$ as in Eq. 5 with $C(\Theta) = 2$. As discussed in Thm. 3, *UCB* and *mUCB* define the boundaries of the performance of *tUCB*. In fact, at the beginning *tUCB* selects arms according to a *UCB* strategy, since no prior information about the models $\Theta$ is available. On the other hand, as more tasks are observed, *tUCB* is able to transfer the knowledge acquired through episodes and build an increasingly accurate estimate of the models, thus approaching the behavior of *mUCB*. This is also confirmed by Fig. 6 where we show how the complexity of *tUCB* changes through episodes. In both cases (regret and complexity) we see that *tUCB* does not reach the same performance of *mUCB*. This is due to the fact that some models have relatively small gaps and thus the number of episodes to have an accurate enough estimate of the models to reach the performance of *mUCB* is much larger than 5000 (see also the Remarks of Thm. 3). Since the final objective is to achieve a small global regret (Eq. 1), in Fig. 7 we report the cumulative regret averaged over the total number of tasks ($J$) for different values of $J$ and $n$. Again, this graph shows that *tUCB* outperforms *UCB* and that it tends to approach the performance of *mUCB* as $J$ increases, for any value of $n$.

## 6  Conclusions and Open Questions

In this paper we introduce the transfer problem in the multi–armed bandit framework when a tasks are drawn from a finite set of bandit problems. We first introduced the bandit algorithm *mUCB* and we showed that it is able to leverage the prior knowledge on the set of bandit problems $\Theta$ and reduce the regret w.r.t. *UCB*. When the set of models is unknown we define a method-of-moments variant (*RTP*) which consistently estimates the means of the models in $\Theta$ from the samples collected through episodes. This knowledge is then transferred to *umUCB* which performs no worse than *UCB* and tends to approach the performance of *mUCB*. For these algorithms we derive regret bounds, and we show preliminary numerical simulations. To the best of our knowledge, this is the first work studying the problem of transfer in multi-armed bandit. It opens a series of interesting directions, including whether explicit model identification can improve our transfer regret.

*Optimality of tUCB*. At each episode, *tUCB* transfers the knowledge about $\Theta$ acquired from previous tasks to achieve a small per-episode regret using *umUCB*. Although this strategy guarantees that the per-episode regret of *tUCB* is never worse than UCB, it may not be the optimal strategy in terms of the cumulative regret through episodes. In fact, if $J$ is large, it could be preferable to run a *model identification* algorithm instead of *umUCB* in earlier episodes so as to improve the quality of the estimates $\hat{\mu}_i(\theta)$. Although such an algorithm would incur a much larger regret in earlier tasks (up to linear), it could approach the performance of *mUCB* in later episodes much faster than done by *tUCB*. This trade-off between *identification* of the models and *transfer* of knowledge may suggest that different algorithms than *tUCB* are possible.

*Unknown model-set size*. In some problems the size of model set $m$ is not known to the learner and needs to be estimated. This problem can be addressed by estimating the rank of matrix $M_2$ which equals to $m$ (Kleibergen and Paap, 2006). We also note that one can relax the assumption that $\rho(\theta)$ needs to be positive (see Assumption 1) by using the estimated model size as opposed to $m$, since $M_2$ depends not on the means of models with $\rho(\theta) = 0$.

**Acknowledgments.** This research was supported by the National Science Foundation (NSF award #SBE-0836012). We would like to thank Sham Kakade and Animashree Anandkumar for valuable discussions. A. Lazaric would like to acknowledge the support of the Ministry of Higher Education and Research, Nord-Pas-de-Calais Regional Council and FEDER through the' Contrat de Projets Etat Region (CPER) 2007-2013', and the European Community's Seventh Framework Programme (FP7/2007-2013) under grant agreement 231495 (project CompLACS).

## Footnotes

[2] Notice that $1/3([\overline{\mu}_{1l}]_i + [\overline{\mu}_{2l}]_i + [\overline{\mu}_{1l}]_i) = \hat{\mu}_{i,n}^l$, the empirical mean of arm $i$ at the end of episode $l$.

[3] The matrix $\widehat{W} \in \mathbb{R}^{K \times m}$ is such that $\widehat{M_2}(\widehat{W}, \widehat{W}) = \mathbf{I}^{m \times m}$, i.e., $\widehat{W}$ is the whitening matrix of $\widehat{M_2}$. In

[4]Notice that although $\Theta$ satisfies Assumption 1, the smallest singular value $\sigma_{\min} = 0.0039$ and $\Gamma_\sigma = 0.0038$, thus making the estimation of the models difficult.

[5] For instance, for *UCB* we compute $\sum_i 1/\Delta_i$.

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
