[Supplementary Material · transfer_fin_supp.pdf]

# Appendices

## A Table of Notation

| Symbol | Explanation |
|---|---|
| $\mathcal{A}$ | Set of arms |
| $\Theta$ | Set of models |
| $K$ | Number of arms |
| $m$ | Number of models |
| $J$ | Number of episodes |
| $n$ | Number of steps per episode |
| $t$ | Time step |
| $\bar{\theta}$ | Current model |
| $\Theta_t$ | Active set of models at time $t$ |
| $\nu_i$ | Distribution of arm $i$ |
| $\mu_i(\theta)$ | Mean of arm $i$ for model $\theta$ |
| $\mu(\theta)$ | Vector of means of model $\theta$ |
| $\hat{\mu}_{i,t}$ | Estimate of $\mu_i(\bar{\theta})$ at time $t$ |
| $\hat{\mu}_i^j(\theta)$ | Estimate of $\mu_i(\theta)$ by RTP for model $\theta$ and arm $i$ at episode $j$ |
| $\widehat{\mu}^j(\theta)$ | Estimate of $\mu(\theta)$ by RTP for model $\theta$ at episode $j$ |
| $\Theta^j$ | Estimated model of RTP after $j$ episode |
| $\varepsilon^j$ | Uncertainty of the estimated model by RTP at episode $j$ |
| $\varepsilon_{i,t}$ | Model uncertainty at time $t$ |
| $\delta$ | Probability of failure |
| $i_*(\theta)$ | Best arm of model $\theta$ |
| $\mu_*(\theta)$ | Optimal value of model $\theta$ |
| $\Delta_i(\theta)$ | Arm gap of an arm $i$ for a model $\theta$ |
| $\Gamma_i(\theta,\theta')$ | Model gap for an arm $i$ between two models $\theta$ and $\theta'$ |
| $M_2$ | $2^{\text{nd}}$-order moment |
| $M_3$ | $3^{\text{rd}}$-order moment |
| $\widehat{M_2}$ | Empirical $2^{\text{nd}}$-order moment |
| $\widehat{M_3}$ | Empirical $3^{\text{rd}}$-order moment |
| $\lVert \cdot \rVert$ | Euclidean norm |
| $\lVert \cdot \rVert_F$ | Frobenius norm |
| $\lVert \cdot \rVert_{\max}$ | Matrix max-norm |
| $\mathcal{R}_J$ | Pseudo-regret |
| $T_{i,n}^j$ | The number of pulls to arm $i$ after $n$ steps of episode $j$ |
| $\mathcal{A}_*(\Theta')$ | Set of arms which are optimal for at least a model in a set $\Theta'$ |
| $\Theta(\mathcal{A}')$ | Set of models for which the arms in $\mathcal{A}'$ are optimal |
| $\Theta_+$ | Set of optimistic models for a given model $\bar{\theta}$ |
| $\mathcal{A}_+$ | Set of optimal arms corresponds to $\Theta_+$ |
| $W$ | Whitening matrix of $M_2$ |
| $\widehat{W}$ | Empirical whitening matrix |
| $T$ | $M_2$ under the linear transformation $W$ |
| $\widehat{T}$ | $\widehat{M_2}$ under the linear transformation $\widehat{W}$ |
| $D$ | Diagonal matrix consisting of the $m$ largest eigenvalues of $M_2$ |
| $\widehat{D}$ | Diagonal matrix consisting of the $m$ largest eigenvalues of $\widehat{M_2}$ |
| $U$ | $K \times m$ matrix with the corresponding eigenvectors of $D$ as its columns |
| $\widehat{U}$ | $K \times m$ matrix with the corresponding eigenvectors of $\widehat{D}$ as its columns |
| $\lambda(\theta)$ | Eigenvalue of $T$ associated with $\theta$ |
| $v(\theta)$ | Eigenvector of $T$ associated with $\theta$ |
| $\widehat{\lambda}(\theta)$ | Eigenvalue of $\widehat{T}$ associated with $\theta$ |
| $\widehat{v}(\theta)$ | Eigenvector of $\widehat{T}$ associated with $\theta$ |
| $\Sigma_{M_2}$ | Set of $m$ largest eigenvalues of the matrix $M_2$ |
| $\sigma_{\min}$ | Minimum eigenvalue of $M_2$ among the $m$-largest |
| $\sigma_{\max}$ | Maximum eigenvalue of $M_2$ |
| $\lambda_{\max}$ | Maximum eigenvalue of $T$ |
| $\Gamma_\sigma$ | Minimum gap between the eigenvalues of $M_2$ |
| $C(\Theta)$ | $O\left(\lambda_{\max}\sqrt{\frac{\sigma_{\max}}{\sigma_{\min}^3}}\left(\frac{\sigma_{\max}}{\Gamma_\sigma} + \frac{1}{\sigma_{\min}} + \frac{1}{\sigma_{\max}}\right)\right)$ |
| $\pi(\theta)$ | Permutation on $\theta$ |
| $\mathcal{A}_*^j(\theta)$ | Set of non-dominated arms for model $\theta$ at episode $j$ |
| $\widetilde{\Theta}_+^j$ | Set of models that cannot be discarded at episode $j$ |
| $\Theta_{i,+}^j$ | Set of models for which $i$ is among the optimistic non-dominated arms at episode $j$ |

# B Proofs of Section 3

**Lemma 1.** *mUCB never pulls arms which are not optimal for at least one model, that is $\forall i \notin \mathcal{A}_*(\Theta)$, $T_{i,n} = 0$ with probability 1. Notice also that $|\mathcal{A}_*(\Theta)| \leq |\Theta|$.*

**Lemma 2.** *The actual model $\bar{\theta}$ is never discarded with high-probability. Formally, the event $\mathcal{E} = \{\forall t = 1, \ldots, n, \bar{\theta} \in \Theta_t\}$ holds with probability $\mathbb{P}[\mathcal{E}] \geq 1 - \delta$ if*

$$\varepsilon_{i,t} = \sqrt{\frac{1}{2T_{i,t-1}} \log\left(\frac{mn^2}{\delta}\right)},$$

*where $T_{i,t-1}$ is the number of pulls to arm $i$ at the beginning of step $t$ and $m = |\Theta|$.*

In the previous lemma we implicitly assumed that $|\Theta| = m \leq K$. In general, the best choice in the definition of $\varepsilon_{i,t}$ has a logarithmic factor with $\min\{|\Theta|, K\}$.

**Lemma 3.** *On event $\mathcal{E}$, all the arms $i \notin \mathcal{A}_*(\Theta_+)$, i.e., arms which are not optimal for any of the optimistic models, are never pulled, i.e., $T_{i,n} = 0$ with probability $1 - \delta$.*

The previous lemma suggests that *mUCB* tends to discard all the models in $\Theta_+$ from the most optimistic down to the actual model $\bar{\theta}$ which, on event $\mathcal{E}$, is never discarded. As a result, even if other models are still in $\Theta_t$, the optimal arm of $\bar{\theta}$ is pulled until the end. Finally, we show that the model gaps of interest (see Thm. 1) are always bigger than the arm gaps.

**Lemma 4.** *For any model $\theta \in \Theta_+$, $\Gamma_{i_*(\theta)}(\theta, \bar{\theta}) \geq \Delta_{i_*(\theta)}(\bar{\theta})$.*

*Proof of Lem. 1.* From the definition of the algorithm we notice that $I_t$ can only correspond to the optimal arm $i_*$ of one model in the set $\Theta_t$. Since $\Theta_t$ can at most contain all the models in $\Theta$, all the arms which are not optimal are never pulled. $\qquad\square$

*Proof of Lem. 2.* We compute the probability of the complementary event $\mathcal{E}^C$, that is that event on which there exist at least one step $t = 1, \ldots, n$ where the true model $\bar{\theta}$ is not in $\Theta_t$. By definition of $\Theta_t$, we have that

$$\mathcal{E} = \{\forall t, \bar{\theta} \in \Theta_t\} = \{\forall t, \forall i \in \mathcal{A}, |\mu_i - \hat{\mu}_{i,t}| \leq \varepsilon_{i,t}\},$$

then

$$\mathbb{P}[\mathcal{E}^C] = \mathbb{P}[\exists t, i, |\mu_i - \hat{\mu}_{i,t}| \geq \varepsilon_{i,t}] \leq \sum_{t=1}^{n} \sum_{i \in \mathcal{A}} \mathbb{P}[|\mu_i - \hat{\mu}_{i,t}| \geq \varepsilon_{i,t}] = \sum_{t=1}^{n} \sum_{i \in \mathcal{A}^*(\Theta)} \mathbb{P}[|\mu_i - \hat{\mu}_{i,t}| \geq \varepsilon_{i,t}]$$

where the upper-bounding is a simple union bound and the last passage comes from the fact that the probability for the arms which are never pulled is always 0 according to Lem. 1. At time $t$, $\hat{\mu}_{i,t}$ is the empirical average of the $T_{i,t-1}$ samples observed from arm $i$ up to the beginning of round $t$. We define the confidence $\varepsilon_{i,t}$ as

$$\varepsilon_{i,t} = \sqrt{\frac{1}{2T_{i,t-1}} \log\left(\frac{|\Theta|n^\alpha}{\delta}\right)},$$

where $\delta \in (0, 1)$ and $\alpha$ is a constant chosen later. Since $T_{i,t-1}$ is a random variable, we need to take an additional union bound over $T_{i,t-1} = 1, \ldots, t - 1$ thus obtaining

$$\mathbb{P}[\mathcal{E}^C] \leq \sum_{t=1}^{n} \sum_{i \in \mathcal{A}^*(\Theta)} \sum_{T_{i,t-1}=1}^{t-1} \mathbb{P}[|\mu_i - \hat{\mu}_{i,t}| \geq \varepsilon_{i,t}]$$

$$\leq \sum_{t=1}^{n} \sum_{i \in \mathcal{A}^*(\Theta)} \sum_{T_{i,t-1}=1}^{t-1} 2 \exp\left(-2T_{i,t-1}\varepsilon_{i,t}^2\right) \leq n(n-1)\frac{|\mathcal{A}^*(\Theta)|\delta}{|\Theta|n^\alpha}.$$

Since $|\mathcal{A}^*(\Theta)| < |\Theta|$ (see Lem. 1) and by taking $\alpha = 2$ we finally have $\mathbb{P}[\mathcal{E}^C] \leq \delta$. $\qquad\square$

*Proof of Lem. 3.* On event $\mathcal{E}$, $\Theta_t$ always contains the true model $\bar{\theta}$, thus only models with larger optimal value could be selected as the optimistic model $\theta_t = \arg\max_{\theta \in \Theta_t} \mu_*(\theta)$, thus restricting the focus of the algorithm only to the models in $\Theta_+$ and their respective optimal arms. $\qquad\square$

*Proof of Lem. 4.* By definition of $\Theta_+$ we have $\mu_{i_*(\theta)}(\theta) = \mu_*(\theta) > \mu_*(\bar{\theta})$ and by definition of optimal arm we have $\mu_*(\bar{\theta}) > \mu_{i_*(\theta)}(\bar{\theta})$, hence $\mu_*(\theta) > \mu_{i_*(\theta)}(\bar{\theta})$. Recalling the definition of model gap, we have $\Gamma_{i_*(\theta)}(\theta) = |\mu_{i_*(\theta)}(\theta) - \mu_{i_*(\theta)}(\bar{\theta})| = \mu_*(\theta) - \mu_{i_*(\theta)}(\bar{\theta})$, where we used the definition of $\mu_*(\theta)$ and the previous inequality. Using the definition of arm gap $\Delta_i$, we obtain

$$\Gamma_{i_*(\theta)}(\theta, \bar{\theta}) = \mu_*(\theta) - \mu_{i_*(\theta)}(\bar{\theta}) \geq \mu_*(\bar{\theta}) - \mu_{i_*(\theta)}(\bar{\theta}) = \Delta_{i_*(\theta)}(\bar{\theta}),$$

which proves the statement. $\qquad\square$

*Proof of Thm. 1.* We decompose the expected regret as

$$\mathbb{E}[\mathcal{R}_n] = \sum_{i \in \mathcal{A}} \Delta_i \mathbb{E}[T_{i,n}] = \sum_{i \in \mathcal{A}_*(\Theta)} \Delta_i \mathbb{E}[T_{i,n}] \leq n\mathbb{P}\{\mathcal{E}^C\} + \sum_{i \in \mathcal{A}_+} \Delta_i \mathbb{E}[T_{i,n}|\mathcal{E}],$$

where the refinement on the sum over arms follows from Lem. 1 and 3 and the high probability event $\mathcal{E}$. In the following we drop the dependency on $\bar{\theta}$ and we write $\mu_i(\bar{\theta}) = \mu_i$.

We now bound the regret when the correct model is always included in $\Theta_t$. On event $\mathcal{E}$, only the restricted set of *optimistic* models $\Theta_+ = \{\theta \in \Theta : \mu_*(\theta) \geq \mu_*\}$ is actually used by the algorithm. Thus we need to compute the number of pulls to the suboptimal arms before all the models in $\Theta_+$ are discarded from $\Theta_t$. We first compute the number of pulls to an arm $i$ needed to discard a model $\theta$ on event $\mathcal{E}$. We notice that

$$\theta \in \Theta_t \Leftrightarrow \{\forall i \in \mathcal{A}, |\mu_i(\theta) - \hat{\mu}_{i,t}| \leq \varepsilon_{i,t}\},$$

which means that a model $\theta$ is included only when all its means are *compatible* with the current estimates. Since we consider event $\mathcal{E}$, $|\mu_i - \hat{\mu}_{i,t}| \leq \varepsilon_{i,t}$, thus $\theta \in \Theta_t$ only if for all $i \in \mathcal{A}$

$$2\varepsilon_{i,t} \geq \Gamma_i(\theta, \bar{\theta}),$$

which corresponds to

$$T_{i,t-1} \leq \frac{2}{\Gamma_i(\theta, \bar{\theta})^2} \log\left(\frac{|\Theta|n^2}{\delta}\right), \tag{6}$$

which implies that if there exists at least one arm $i$ for which at time $t$ the number of pulls $T_{i,t}$ exceeds the previous quantity, then $\forall s > t$ we have $\theta \notin \Theta_t$ (with probability $\mathbb{P}(\mathcal{E})$). To obtain the final bound on the regret, we recall that the algorithm first selects an optimistic model $\theta_t$ and then it pulls the corresponding optimal arm until the optimistic model is not discarded. Thus we need to compute the number of times the optimal arm of the optimistic model is pulled before the model is discarded. More formally, since we know that on event $\mathcal{E}$ we have that $T_{i,n} = 0$ for all $i \notin \mathcal{A}_+$, the constraints of type (6) could only be applied to the arms $i \in \mathcal{A}_+$. Let $t$ be the last time arm $i$ is pulled, which coincides, by definition of the algorithm, with the last time any of the models in $\Theta_{+,i} = \{\theta \in \Theta_+ : i_*(\theta) = i\}$ (i.e., the optimistic models recommending $i$ as the optimal arm) is included in $\Theta_t$. Then we have that $T_{i,t-1} = T_{i,n} - 1$ and the fact that $i$ is pulled corresponds to the fact the a model $\theta_i \in \Theta_{+,i}$ is such that

$$\theta_i \in \Theta_t \wedge \forall \theta' \in \Theta_t, \mu_*(\theta_i) > \mu_*(\theta'),$$

which implies that (see Eq. 6)

$$T_{i,n} \leq \frac{2}{\min_{\theta \in \Theta_{+,i}} \Gamma_i(\theta, \bar{\theta})^2} \log\left(\frac{|\Theta|n^2}{\delta}\right) + 1. \tag{7}$$

where the minimum over $\Theta_{+,i}$ guarantees that all the optimistic models with optimal arm $i$ are actually discarded.

Grouping all the conditions, we obtain the expected regret

$$\mathbb{E}[\mathcal{R}_n] \leq K + \sum_{i \in \mathcal{A}_+} \frac{2\Delta_i(\bar{\theta})}{\min_{\theta \in \Theta_{+,i}} \Gamma_i(\theta, \bar{\theta})^2} \log\left(|\Theta|n^3\right)$$

with $\delta = 1/n$. Finally we can apply Lem. 4 which guarantees that for any $\theta \in \Theta_{+,i}$ the gaps $\Gamma_i(\theta, \bar{\theta}) \geq \Delta_i(\bar{\theta})$ and obtain the final statement. $\qquad\square$

**Remark (proof).** The proof of the theorem considers a worst case. In fact, while pulling the optimal arm of the optimistic model $i_*(\theta_t)$ we do not consider that the algorithm might actually discard other models, thus reducing $\Theta_t$ before the optimistic model is actually discarded. More formally, we assume that for any $\theta \in \Theta_t$ not in $\Theta_{+,i}$ the number of steps needed to be discarded by pulling $i_*(\theta_t)$ is larger than the number of pulls needed to discard $\theta_t$ itself, which corresponds to

$$\min_{\theta \in \Theta_{+,i}} \Gamma_i^2(\theta, \bar\theta) \geq \max_{\substack{\theta \in \Theta^+ \\ \theta \notin \Theta_{+,i}}} \Gamma_i^2(\theta, \bar\theta).$$

Whenever this condition is not satisfied, the analysis is suboptimal since it does not fully exploit the structure of the problem and *mUCB* is expected to perform better than predicted by the bound.

**Remark (comparison to *UCB* with hypothesis testing).** An alternative strategy is to pair *UCB* with hypothesis testing of fixed confidence $\delta$. Let $\Gamma_{\min}(\bar\theta) = \min_i \min_\theta \Gamma_i(\theta, \bar\theta)$, if at time $t$ there exists an arm $i$ such that $T_{i,t} > 2\log(2/\delta)\Gamma_{\min}^2$, then all the models $\theta \neq \bar\theta$ can be discarded with probability $1 - \delta$. Since from the point of view of the hypothesis testing the exploration strategy is unknown, we can only assume that after $\tau$ steps we have $T_{i,\tau} \geq \tau/K$ for at least one arm $i$. Thus after $\tau > 2K\log(2/\delta)/\Gamma_{\min}^2$ steps, the hypothesis testing returns a model $\hat\theta$ which coincides with $\bar\theta$ with probability $1 - \delta$. If $\tau \leq n$, from time $\tau$ on, the algorithm always pulls $I_t = i_*(\hat\theta)$ and incurs a zero regret with high probability. If we assume $\tau \leq n$, the expected regret is

$$\mathbb{E}[\mathcal{R}_n(\text{UCB+Hyp})] \leq O\bigg( \sum_{i \in \mathcal{A}} \frac{\log n\tau}{\Delta_i} \bigg) \leq O\bigg( K\frac{\log n\tau}{\Delta} \bigg).$$

We notice that this algorithm only has a mild improvement w.r.t. standard *UCB*. In fact, in *UCB* the big-$O$ notation hides the constants corresponding to the exponent of $n$ in the logarithmic term. This suggests that whenever $\tau$ is much smaller than $n$, then there might be a significant improvement. On the other hand, since $\tau$ has an inverse dependency w.r.t. $\Gamma_{\min}$, it is very easy to build model sets $\Theta$ where $\Gamma_{\min} = 0$ and obtain an algorithm with exactly the same performance as *UCB*.

## C Sample Complexity Analysis of *RTP*

In this section we provide the full sample complexity analysis of the *RTP* algorithm. In our analysis we rely on some results of Anandkumar et al. (2012b). Anandkumar et al. (2012b) have provided perturbation bounds on the error of the orthonormal eigenvectors $\hat{v}(\theta)$ and the corresponding eigenvalues $\hat\lambda(\theta)$ in terms of the perturbation error of the transformed tensor $\epsilon = \|T - \hat{T}\|$ (see Anandkumar et al., 2012b, Thm 5.1). However, this result does not provide us with the sample complexity bound on the estimation error of model means. Here we complete their analysis by proving a sample complexity bound on the $\ell_2$-norm of the estimation error of the means $\|\mu(\theta) - \hat\mu(\theta)\|$.

We follow the following steps in our proof: **(i)** we bound the error $\epsilon$ in terms of the estimation errors $\epsilon_2 := \|\widehat{M}_2 - M_2\|$ and $\epsilon_3 := \|\widehat{M}_3 - M_3\|$ (Lem. 6). **(ii)** we prove high probability bounds on the error $\epsilon_2$ and $\epsilon_3$ using some standard concentration inequality results (Lem. 7). The bounds on the errors of the estimates $\hat{v}(\theta)$ and $\hat\lambda(\theta)$ immediately follow from combining the results of Lem. 6, Lem. 7 and Thm. 5. **(iii)** Based on these bounds we then prove our main result by bounding the estimation error associated with the inverse transformation $\hat\mu(\theta) = \hat\lambda(\theta)\widehat{B}\hat{v}(\theta)$ in high probability.

We begin by recalling the perturbation bound of Anandkumar et al. (2012b):

**Theorem 5** (Anandkumar et al., 2012b)**.** *Pick $\eta \in (0, 1)$. Define $W := UD^{-1/2}$, where $D \in \mathbb{R}^{m \times m}$ is the diagonal matrix of the $m$ largest eigenvalues of $M_2$ and $U \in \mathbb{R}^{K \times m}$ is the matrix with the eigenvectors associated with the $m$ largest eigenvalues of $M_2$ as its columns. Then $W$ is a linear mapping which satisfies $W^\mathsf{T} M_2 W = \mathbf{I}$. Let $\hat{T} = T + E \in \mathbb{R}^{m \times m \times m}$, where the $3^{\mathrm{rd}}$ order moment tensor $T = M_3(W, W, W)$ is symmetric and orthogonally decomposable in the form of $\sum_{\theta \in \Theta} \lambda(\theta)v(\theta)^{\otimes 3}$, where each $\lambda(\theta) > 0$ and $\{v(\theta)\}_\theta$ is an orthonormal basis. Define $\epsilon := \|E\|$ and $\lambda_{\max} = \max_\theta \lambda(\theta)$. Then there exist some constants $C_1, C_2 > 0$, some polynomial function $f(\cdot)$, and a permutation $\pi$ on $\Theta$ such that the following holds w.p. $1 - \eta$*

$$\|v(\theta) - \hat{v}(\pi(\theta))\| \leq 8\epsilon/\lambda(\theta),$$

$$|\lambda(\theta) - \hat\lambda(\pi(\theta))| \leq 5\epsilon,$$

*for $\epsilon \leq C_1 \frac{\lambda_{\min}}{m}$, $L > \log(1/\eta) f(k)$ and $N \geq C_2(\log(k) + \log\log(\lambda_{\max}/\epsilon))$, where $N$ and $L$ are the internal parameters of RTP algorithm.*

For ease of exposition we consider the *RTP* algorithm in asymptotic case, i.e., $N, L \to \infty$ and $\eta \approx 1$. We now prove bounds on the perturbation error $\epsilon$ in terms of the estimation error $\epsilon_2$ and $\epsilon_3$. This requires bounding the error between $W = UD^{-1/2}$ and $\widehat{W} = \widehat{U}\widehat{D}^{-1/2}$ using the following perturbation bounds on $\|U - \widehat{U}\|$, $\|\widehat{D}^{-1/2} - D^{-1/2}\|$ and $\|\widehat{D}^{1/2} - D^{1/2}\|$.

**Lemma 5.** *Assume that $\epsilon_2 \leq 1/2 \min(\Gamma_\sigma, \sigma_{\min})$, then we have*

$$\|\widehat{D}^{-1/2} - D^{-1/2}\| \leq \frac{2\epsilon_2}{(\sigma_{\min})^{3/2}}, \quad \text{and} \quad \|\widehat{D}^{1/2} - D^{1/2}\| \leq \frac{\epsilon_2}{\sigma_{\max}}, \quad \text{and} \quad \|\widehat{U} - U\| \leq \frac{2\sqrt{m}\epsilon_2}{\Gamma_\sigma}.$$

*Proof.* Here we just prove bounds on $\|\widehat{D}^{-1/2} - D^{-1/2}\|$ and $\|\widehat{U} - U\|$. The bound on $\|\widehat{D}^{-1/2} - D^{-1/2}\|$ can be proven using a similar argument to that used for bounding $\|\widehat{D}^{1/2} - D^{1/2}\|$. Let $\widehat{\Sigma}_m = \{\widehat{\sigma}_1, \widehat{\sigma}_2, \ldots, \widehat{\sigma}_m\}$ be the set of $m$ largest eigenvalues of the matrix $\widehat{M}_2$. We have

$$\|\widehat{D}^{-1/2} - D^{-1/2}\| \overset{(1)}{=} \max_{1 \leq i \leq m} \left| \sqrt{\frac{1}{\sigma_i}} - \sqrt{\frac{1}{\widehat{\sigma}_i}} \right| = \max_{1 \leq i \leq m} \left( \frac{\left| \frac{1}{\sigma_i} - \frac{1}{\widehat{\sigma}_i} \right|}{\sqrt{\frac{1}{\sigma_i}} + \sqrt{\frac{1}{\widehat{\sigma}_i}}} \right)$$

$$\leq \max_{1 \leq i \leq m} \left( \sqrt{\sigma_i} \left| \frac{1}{\sigma_i} - \frac{1}{\widehat{\sigma}_i} \right| \right) \leq \max_{1 \leq i \leq m} \left| \frac{\sigma_i - \widehat{\sigma}_i}{\sqrt{\sigma_i \widehat{\sigma}_i}} \right| \overset{(2)}{\leq} \frac{\epsilon_2}{\sqrt{\sigma_{\min}}(\sigma_{\min} - \epsilon_2)} \overset{(3)}{\leq} \frac{2\epsilon_2}{(\sigma_{\min})^{3/2}},$$

where in (1) we use the fact that the spectral norm of matrix is its largest singular value, which in case of a diagonal matrix coincides with its biggest element, in (2) we rely on the result of Weyl (see Stewart and Sun, 1990, Thm. 4.11, p. 204) for bounding the difference between $\sigma_i$ and $\widehat{\sigma}_i$, and in (3) we make use of the assumption that $\epsilon_2 \leq 1/2\sigma_{\min}$.

In the case of $\|U - \widehat{U}\|$ we rely on the perturbation bound of Wedin (1972). This result guarantees that for any positive definite matrix $A$ the difference between the eigenvectors of $A$ and the perturbed $\widehat{A}$ (also positive definite) is small whenever there is a minimum gap between the eigenvalues of $\widehat{A}$ and $A$. More precisely, for any positive definite matrix $A$ and $\widehat{A}$ such that $||A - \widehat{A}|| \leq \epsilon_A$, let the minimum eigengap be $\Gamma_{A \leftrightarrow \widehat{A}} := \min_{j \neq i} |\sigma_i - \widehat{\sigma}_j|$, then we have

$$\|u_i - \widehat{u}_i\| \leq \frac{\epsilon_A}{\Gamma_{A \leftrightarrow \widehat{A}}}, \tag{8}$$

where $(u_i, \sigma_i)$ is an eigenvalue/vector pair for the matrix $A$. Based on this result we now bound the error $\|U - \widehat{U}\|$

$$\|U - \widehat{U}\| \leq \|U - \widehat{U}\|_F \leq \sqrt{\sum_i \|u_i - \widehat{u}_i\|^2} \overset{(1)}{\leq} \frac{\sqrt{m}\epsilon_2}{\Gamma_{M_2 \leftrightarrow \widehat{M}_2}} \overset{(2)}{\leq} \frac{\sqrt{m}\epsilon_2}{\Gamma_\sigma - \epsilon_2} \overset{(3)}{\leq} \frac{2\sqrt{m}\epsilon_2}{\Gamma_\sigma},$$

where in (1) we rely on Eq. 8 and in (2) we rely on the definition of the gap as well as Weyl's inequality. Finally, in (3) We rely on the fact that $\epsilon_2 \leq 1/2\Gamma_\sigma$ for bounding denominator from below.

Our result also holds for those cases where the multiplicity of some of the eigenvalues are greater than 1. Note that for any eigenvalue $\lambda$ with multiplicity $l$ the linear combination of the corresponding eigenvectors $\{v_1, v_2, \ldots, v_l\}$ is also an eigenvector of the matrix. Therefore, in this case it suffices to bound the difference between the eigenspaces of two matrix. The result of Wedin (1972) again applies to this case and bounds the difference between the eigenspaces in terms of the perturbation $\epsilon_2$ and $\Gamma_\sigma$. $\square$

We now bound $\epsilon$ in terms of $\epsilon_2$ and $\epsilon_3$.

**Lemma 6.** *Let $\mu_{\max} := \max_\theta \|\mu(\theta)\|$, if $\epsilon_2 \leq 1/2 \min(\Gamma_\sigma, \sigma_{\min})$, then the estimation error $\epsilon$ is bounded as*

$$\epsilon \leq \left( \frac{m}{\sigma_{\min}} \right)^{3/2} \left( 10\epsilon_2 \left( \frac{1}{\Gamma_\sigma} + \frac{1}{\sigma_{\min}} \right) (\epsilon_3 + \mu_{\max}^3) + \epsilon_3 \right).$$

*Proof.* Based on the definitions of $T$ and $\widehat{T}$ we have

$$\epsilon = \|T - \widehat{T}\| = \|M_3(W, W, W) - \widehat{M}_3(\widehat{W}, \widehat{W}, \widehat{W})\|$$

$$\leq \|M_3(W, W, W) - \widehat{M}_3(W, W, W)\| + \|\widehat{M}_3(W, W, W) - \widehat{M}_3(W, W, \widehat{W})\|$$

$$+ \|\widehat{M}_3(W, W, \widehat{W}) - \widehat{M}_3(W, \widehat{W}, \widehat{W})\| + \|\widehat{M}_3(W, \widehat{W}, \widehat{W}) - \widehat{M}_3(\widehat{W}, \widehat{W}, \widehat{W})\| \qquad (9)$$

$$= \|E_{M_3}(W, W, W)\| + \|\widehat{M}_3(W, W, W - \widehat{W})\| + \|\widehat{M}_3(W, W - \widehat{W}, \widehat{W})\|$$

$$+ \|\widehat{M}_3(W - \widehat{W}, \widehat{W}, \widehat{W})\|,$$

where $E_{M_3} = M_3 - \widehat{M}_3$. We now bound the terms in the r.h.s. of Eq. 9 in terms of $\epsilon_3$ and $\epsilon_2$. We begin by bounding $\|E_{M_3}(W, W, W)\|$:

$$\|E_{M_3}(W, W, W)\| \leq \|E_{M_3}\|\|W\|^3 \leq \|E_{M_3}\|\|U\|^3\|D^{-1}\|^{3/2} \leq \|E_{M_3}\|\|U\|_F^3\|D^{-1}\|^{3/2}$$

$$\overset{(1)}{=} \left(\frac{m}{\sigma_{\min}}\right)^{3/2} \|E_{M_3}\| \leq \left(\frac{m}{\sigma_{\min}}\right)^{3/2} \epsilon_3, \qquad (10)$$

where in (1) we use the fact that $U$ is an orthonormal matrix and $D$ is diagonal. In the case of $\|\widehat{M}_3(W, W, W - \widehat{W})\|$ we have

$$\|\widehat{M}_3(W, W, W - \widehat{W})\| \leq \|W\|^2\|W - \widehat{W}\|\|\widehat{M}_3\| \leq \|W\|^2\|W - \widehat{W}\|(\|\widehat{M}_3 - M_3\| + \|M_3\|)$$

$$\overset{(1)}{\leq} \|W\|^2\|W - \widehat{W}\|(\epsilon_3 + \mu_{\max}^3) \leq \|W\|^2\|UD^{-1/2} - \widehat{U}\widehat{D}^{-1/2}\|(\epsilon_3 + \mu_{\max}^3)$$

$$\leq \|W\|^2(\|(U - \widehat{U})D^{-1/2}\| + \|\widehat{U}(\widehat{D}^{-1/2} - D^{-1/2})\|)\left(\epsilon_3 + \mu_{\max}^3\right)$$

$$\leq \|W\|^2 \left(\frac{\|U - \widehat{U}\|}{\sqrt{\sigma_{\min}}} + \sqrt{m}\|\widehat{D}^{-1/2} - D^{-1/2}\|\right)\left(\epsilon_3 + \mu_{\max}^3\right).$$

where in (1) we use the definition of $M_3$ as a linear combination of the tensor product of the means $\mu(\theta)$. This result combined with the result of Lem. 5 and the fact that $\|W\| \leq \sqrt{m/\sigma_{\min}}$ (see Eq. 10) implies that

$$\|\widehat{M}_3(W, W, W - \widehat{W})\| \leq \frac{m}{\sigma_{\min}} \left(\frac{2\sqrt{m}\epsilon_2}{\Gamma_\sigma\sqrt{\sigma_{\min}}} + \frac{2\sqrt{m}\epsilon_2}{(\sigma_{\min})^{3/2}}\right)\left(\epsilon_3 + \mu_{\max}^3\right)$$

$$\leq 2\epsilon_2 \left(\frac{m}{\sigma_{\min}}\right)^{3/2} \left(\frac{1}{\Gamma_\sigma} + \frac{1}{\sigma_{\min}}\right)\left(\epsilon_3 + \mu_{\max}^3\right). \qquad (11)$$

Likewise one can prove the following perturbation bounds for $\widehat{M}_3(W, W - \widehat{W}, \widehat{W})$ and $\widehat{M}_3(W - \widehat{W}, \widehat{W}, \widehat{W})$:

$$\|\widehat{M}_3(W, W - \widehat{W}, \widehat{W})\| \leq 2\sqrt{2}\epsilon_2 \left(\frac{m}{\sigma_{\min}}\right)^{3/2} \left(\frac{1}{\Gamma_\sigma} + \frac{1}{\sigma_{\min}}\right)\left(\epsilon_3 + \mu_{\max}^3\right)$$

$$\|\widehat{M}_3(W - \widehat{W}, \widehat{W}, \widehat{W})\| \leq 4\epsilon_2 \left(\frac{m}{\sigma_{\min}}\right)^{3/2} \left(\frac{1}{\Gamma_\sigma} + \frac{1}{\sigma_{\min}}\right)\left(\epsilon_3 + \mu_{\max}^3\right). \qquad (12)$$

The result then follows by plugging the bounds of Eq. 10, Eq. 11 and Eq. 12 into Eq. 9. $\qquad \square$

We now prove high-probability bounds on $\epsilon_3$ and $\epsilon_2$ when $M_2$ and $M_3$ are estimated by sampling.

**Lemma 7.** *For any $\delta \in (0, 1)$, if $\widehat{M}_2$ and $\widehat{M}_3$ are computed with samples from $j$ episodes, then we that with probability $1 - \delta$:*

$$\epsilon_3 \leq K^{1.5}\sqrt{\frac{6\log(2K/\delta)}{j}} \qquad and \qquad \epsilon_2 \leq 2K\sqrt{\frac{\log(2K/\delta)}{j}}.$$

*Proof.* Using some norm inequalities for the tensors we obtain

$$\epsilon_3 = \|M_3 - \widehat{M}_3\| \leq K^{1.5}\|M_3 - \widehat{M}_3\|_{\max} = K^{1.5}\max_{i,j,x}|[M_3]_{i,j,x} - [\widehat{M}_3]_{i,j,x}|.$$

A similar argument leads to the bound of $K \max_{i,j} |[M_2]_{i,j} - [\widehat{M}_2]_{i,j}|$ on $\epsilon_2$. One can easily show that, for every $1 \leq i, j, x \leq K$, the term $[M_3]_{i,j,x} - [\widehat{M}_3]_{i,j,x}$ and $[M_3]_{i,j,x} - [\widehat{M}_3]_{i,j,x}$ can be expressed as a sum of martingale differences with the maximum value $1/j$. The result then follows by applying the Azuma's inequality (e.g., see Cesa-Bianchi and Lugosi, 2006, appendix, pg. 361) and taking the union bound. □

We now draw our attention to the proof of our main result.

***Proof of Thm. 2.*** We begin by deriving the condition of Eq. 5. The assumption on $\epsilon_2$ in Lem. 6 and the result of Lem. 7 hold at the same time, w.p. $1 - \delta$, if the following inequality holds

$$2K\sqrt{\frac{\log(2K/\delta)}{j}} \leq 1/2 \min(\Gamma_\sigma, \sigma_{\min}).$$

By solving the bound w.r.t. $j$ we obtain

$$j \geq \frac{16K^2 \log(2K/\delta)}{\min(\Gamma_\sigma, \sigma_{\min})^2}. \tag{13}$$

A similar argument applies in the case of the assumption on $\epsilon$ in Thm. 5. The results of Thm. 5 and Lem. 6 hold at the same time if we have

$$\varepsilon \leq \left(\frac{mK}{\sigma_{\min}}\right)^{3/2} \left(20\epsilon_2 \left(\frac{1}{\Gamma_\sigma} + \frac{1}{\sigma_{\min}}\right) + \epsilon_3\right) \leq C_1 \frac{\lambda_{\min}}{m},$$

where in the first inequality we used that $\varepsilon_3 \leq K^{3/2}$ and $\mu_{\max}^3 \leq K^{3/2}$ by their respective definitions. This combined with high probability bounds of Lem. 7 on $\epsilon_1$ and $\epsilon_2$ implies

$$\left(\frac{m}{\sigma_{\min}}\right)^{1.5} \left(20K^{2.5}\sqrt{\frac{\log(4K/\delta)}{j}} \left(\frac{1}{\Gamma_\sigma} + \frac{1}{\sigma_{\min}}\right) + K^{1.5}\sqrt{\frac{6\log(4K/\delta)}{j}}\right) \leq C_1 \frac{\lambda_{\min}}{m}.$$

By solving this bound w.r.t. $j$ (and some simplifications) we obtain w.p. $1 - \delta$

$$j \geq \frac{43^2 m^5 K^6 \log(4K/\delta)}{C_1 \sigma_{\min}^3 \lambda_{\min}^2} \left(\frac{1}{\Gamma_\sigma} + \frac{1}{\sigma_{\min}}\right)^2.$$

Combining this result with that of Eq.13 and taking the union bound leads to the bound of Eq. 5 on the minimum number of samples.

We now draw our attention to the main result of the theorem. We begin by bounding $\|\mu(\theta) - \widehat{\mu}(\pi(\theta))\|$ in terms of estimation error term $\epsilon_3$ and $\epsilon_2$:

$$\|\mu(\theta) - \widehat{\mu}(\pi(\theta))\| = \|\lambda(\theta)Bv(\theta) - \widehat{\lambda}(\pi(\theta))\widehat{B}\widehat{v}(\pi(\theta))\|$$

$$\leq \|(\lambda(\pi(\theta)) - \widehat{\lambda}(\theta))Bv(\pi(\theta))\| + \|\widehat{\lambda}(\theta)(B - \widehat{B})v(\pi(\theta))\| + \|\widehat{\lambda}(\theta)\widehat{B}(v(\pi(\theta)) - \widehat{v}(\theta))\| \tag{14}$$

$$\leq |\lambda(\theta) - \widehat{\lambda}(\pi(\theta))|\|B\| + \widehat{\lambda}(\pi(\theta))\|B - \widehat{B}\| + \widehat{\lambda}(\pi(\theta))\|\widehat{B}\|\|v(\theta) - \widehat{v}(\pi(\theta))\|,$$

where in the last line we rely on the fact that both $v(\theta)$ and $\widehat{v}(\pi(\theta))$ are normalized vectors. We first bound the term $\|B - \widehat{B}\|$:

$$\|B - \widehat{B}\| = \|UD^{1/2} - \widehat{U}\widehat{D}^{1/2}\| \leq \|(U - \widehat{U})D^{1/2}\| + \|\widehat{U}(D^{1/2} - \widehat{D}^{1/2})\|$$

$$\overset{(1)}{\leq} \frac{2\sqrt{m}\epsilon_2 \sigma_{\max}}{\Gamma_\sigma} + \frac{\sqrt{m}\epsilon_2}{\sigma_{\max}} \leq \sqrt{m}\epsilon_2 \left(\frac{2\sigma_{\max}}{\Gamma_\sigma} + \frac{1}{\sigma_{\max}}\right),$$

where in (1) we make use of the result of Lem. 5. Furthermore, we have

$$\|\widehat{B}\| = \|\widehat{U}\widehat{D}^{1/2}\| \leq \sqrt{m\widehat{\sigma}_{\max}} \leq \sqrt{m}(\sigma_{\max}^{1/2} + \epsilon_2^{1/2}) \leq \sqrt{m}(\sigma_{\max}^{1/2} + \sigma_{\min}^{1/2}) \leq \sqrt{2m\sigma_{\max}},$$

where we used the condition on $\epsilon_2$. This combined with Eq.14 and the result of Thm 5 and Lem. 6 implies

$$\|\mu(\pi(\theta)) - \widehat{\mu}(\theta)\|$$

$$\overset{(1)}{\leq} 5\sqrt{m\sigma_{\max}}\epsilon + \sqrt{m}\epsilon_2\left(\lambda(\theta) + \epsilon\right)\left(\frac{2\sigma_{\max}}{\Gamma_\sigma} + \frac{1}{\sigma_{\max}}\right) + \frac{8\epsilon}{\lambda(\theta)}\sqrt{2m\sigma_{\max}}\left(\lambda(\theta) + \epsilon\right)$$

$$\overset{(2)}{\leq} 5\sqrt{m\sigma_{\max}}\epsilon + \sqrt{m}\epsilon_2\left(\lambda(\theta) + 5C_1\frac{\sigma_{\min}}{m}\right)\left(\frac{2\sigma_{\max}}{\Gamma_\sigma} + \frac{1}{\sigma_{\max}}\right) + 8\sqrt{2m\sigma_{\max}}\left(1 + 5C_1\frac{\sigma_{\min}}{m}\right)\epsilon$$

$$\leq 5\sqrt{m\sigma_{\max}}\left(\frac{m}{\sigma_{\min}}\right)^{3/2}\left(10\epsilon_2\left(\frac{1}{\Gamma_\sigma} + \frac{1}{\sigma_{\min}}\right)(\epsilon_3 + \mu_{\max}^3) + \epsilon_3\right)$$

$$+ \sqrt{m}\epsilon_2\left(\lambda(\theta) + 5C_1\frac{\sigma_{\min}}{m}\right)\left(\frac{2\sigma_{\max}}{\Gamma_\sigma} + \frac{1}{\sigma_{\max}}\right)$$

$$+ 8\sqrt{2m\sigma_{\max}}\left(1 + \frac{5C_1}{m}\right)\left(\frac{m}{\sigma_{\min}}\right)^{3/2}\left(10\epsilon_2\left(\frac{1}{\Gamma_\sigma} + \frac{1}{\sigma_{\min}}\right)(\epsilon_3 + \mu_{\max}^3) + \epsilon_3\right).$$

where in (1) we used $\|B\| \leq \sqrt{m\sigma_{\max}}$, the bound on $\widehat{\lambda}(\pi(\theta)) \leq \lambda(\theta) + 5\epsilon$, $\|v(\theta) - \widehat{v}(\pi(\theta))\| \leq 8\epsilon/\lambda(\theta)$, in (2) we used $\lambda(\theta) = 1/\sqrt{\rho(\theta)} \geq 1$ and the condition that $\varepsilon \leq 5C_1\sigma_{\min}/m$. The result then follows by combining this bound with the high probability bound of Lem. 7 and taking union bound as well as collecting the terms. $\square$

## D  Proofs of Section 4.3

**Lemma 8.** *At episode $j$, the arms $i \notin \mathcal{A}_*^j(\Theta; \bar{\theta}^j)$ are never pulled, i.e., $T_{i,n} = 0$.*

**Lemma 9.** *If umUCB is run with*

$$\varepsilon_{i,t} = \sqrt{\frac{1}{2T_{i,t-1}}\log\left(\frac{2mKn^2}{\delta}\right)}, \qquad \varepsilon^j = C(\Theta)\sqrt{\frac{1}{j}\log\left(\frac{2mKJ}{\delta}\right)}, \tag{15}$$

*where $C(\Theta)$ is defined in Thm. 2, then the event $\mathcal{E} = \mathcal{E}_1 \cap \mathcal{E}_2$ is such that $\mathbb{P}[\mathcal{E}] \geq 1 - \delta$ where $\mathcal{E}_1 = \{\forall \theta, t, i, |\hat{\mu}_{i,t} - \mu_i(\theta)| \leq \varepsilon_{i,t}\}$ and $\mathcal{E}_2 = \{\forall j, \theta, i, |\hat{\mu}_i^j(\theta) - \mu_i(\theta)| \leq \varepsilon^j\}$.*

Notice that the event $\mathcal{E}$ implies that for any episode $j$ and step $t$, the actual model is always in the active set, i.e., $\bar{\theta}^j \in \Theta_t^j$.

**Lemma 10.** *At episode $j$, all the arms $i \notin \mathcal{A}_+^j(\Theta_+^j(\bar{\theta}^j); \bar{\theta}^j)$ are never pulled on event $\mathcal{E}$, i.e., $T_{i,n} = 0$ with probability $1 - \delta$.*

**Lemma 11.** *At episode $j$, the arms $i \in \mathcal{A}_+^j(\Theta_+^j(\bar{\theta}^j); \bar{\theta}^j)$ are never pulled more than with a UCB strategy, i.e.,*

$$T_{i,n}^j \leq \frac{2}{\Delta_i(\bar{\theta}^j)^2}\log\left(\frac{2mKn^2}{\delta}\right) + 1, \tag{16}$$

*with probability $1 - \delta$.*

Notice that for *UCB* the logarithmic term in the previous statement would be $\log(Kn^2/\delta)$ which would represent a negligible constant fraction improvement w.r.t. *umUCB* whenever the number of models is of the same order of the number of arms.

**Lemma 12.** *At episode $j$, for any model $\theta \in (\Theta_+^j(\bar{\theta}^j) - \widetilde{\Theta}^j(\bar{\theta}^j))$ (i.e., an optimistic model that can be discarded), the number of pulls to any arm $i \in \mathcal{A}_+^j(\theta; \bar{\theta}^j)$ needed before discarding $\theta$ is*

$$T_{i,n}^j \leq \frac{1}{2\left(\Gamma_i(\theta, \bar{\theta}^j)/2 - \varepsilon^j\right)^2}\log\left(\frac{2mKn^2}{\delta}\right) + 1, \tag{17}$$

*with probability $1 - \delta$.*

*Proof of Lem. 8.* We first notice that the algorithm only pulls arms recommended by a model $\theta \in \Theta_t^j$. Let $\hat{i}_*(\theta) = \arg\max_i B_t^j(i; \theta)$ with $\theta \in \Theta_t^j$, and $i \in \mathcal{A}_*^j(\theta; \bar{\theta}^j)$. According to the selection process, we have

$$B_t^j(i; \theta) < B_t^j(\hat{i}_*; \theta).$$

Since $\theta \in \Theta_t^j$ we have that for any $i$, $|\hat{\mu}_{i,t} - \hat{\mu}_i^j(\theta)| \leq \varepsilon_{i,t} + \varepsilon^j$ which leads to $\hat{\mu}_i^j(\theta) - \varepsilon^j \leq \hat{\mu}_{i,t} + \varepsilon_{i,t}$. Since $\hat{\mu}_i^j(\theta) - \varepsilon^j \leq \hat{\mu}_i^j(\theta) + \varepsilon^j$, then we have that

$$\hat{\mu}_i^j(\theta) - \varepsilon^j \leq \min\{\hat{\mu}_{i,t} + \varepsilon_{i,t}, \hat{\mu}_i^j(\theta) + \varepsilon^j\} = B_t^j(i; \theta).$$

Furthermore from the definition of the $B$-values we deduce that

$$B_t^j(\hat{i}_*; \theta) \leq \hat{\mu}_{\hat{i}_*}^j(\theta) + \varepsilon^j.$$

Bringing together the previous inequalities, we obtain

$$\hat{\mu}_i^j(\theta) - \varepsilon^j \leq \hat{\mu}_{\hat{i}_*}^j(\theta) + \varepsilon^j.$$

which is a contradiction with the definition of non-dominated arms $\mathcal{A}_*^j(\Theta; \bar{\theta}^j)$. $\qquad\square$

*Proof of Lem. 9.* The probability of $\mathcal{E}_1$ is computed in Lem. 2 with the difference that now we need an extra union bound over all the models and that the union bound over the arms cannot be restricted to the number of models. The probability of $\mathcal{E}_2$ follows from Thm. 2. $\qquad\square$

*Proof of Lem. 10.* We first recall that on event $\mathcal{E}$, at any episode $j$, the actual model $\bar{\theta}^j$ is always in the active set $\Theta_t^j$. If an arm $i$ is pulled, then according to the selection strategy, there exists a model $\theta \in \Theta_t$ such that

$$B_t^j(i; \theta) \geq B_t^j(\hat{i}_*(\bar{\theta}^j); \bar{\theta}^j).$$

Since $\hat{i}_*(\bar{\theta}^j) = \arg\max_i B_t^j(i; \bar{\theta}^j)$, then $B_t^j(\hat{i}_*(\bar{\theta}^j); \bar{\theta}^j) \geq B_t^j(i_*(\bar{\theta}^j); \bar{\theta}^j)$ where $i_*(\bar{\theta}^j)$ is the true optimal arm of $\bar{\theta}^j$. By definition of $B(i; \theta)$, on event $\mathcal{E}$ we have that $B_t^j(i_*(\bar{\theta}^j); \bar{\theta}^j) \geq \mu_*(\bar{\theta}^j)$ and that $B_t^j(i; \theta) \leq \hat{\mu}_i^j(\theta) + \varepsilon^j$. Grouping these inequalities we obtain

$$\hat{\mu}_i^j(\theta) + \varepsilon^j \geq \mu_*(\bar{\theta}^j),$$

which, together with Lem. 8, implies that $i \in \mathcal{A}_+^j(\theta; \bar{\theta}^j)$ and that this set is not empty, which corresponds to $\theta \in \Theta_+^j(\bar{\theta}^j)$. $\qquad\square$

*Proof of Lem. 11.* Let $t$ be the last time arm $i$ is pulled ($T_{i,t-1} = T_{i,n} + 1$), then according to the selection strategy we have

$$B_t^j(i; \theta_t^j) \geq B_t^j(\hat{i}_*(\bar{\theta}^j); \bar{\theta}^j) \geq B_t^j(i_*; \bar{\theta}^j),$$

where $i_* = i_*(\bar{\theta}^j)$. Using the definition of $B$, we have that on event $\mathcal{E}$

$$B_t^j(i_*(\bar{\theta}^j); \bar{\theta}^j) = \min\left\{(\hat{\mu}_{i_*}^j(\bar{\theta}^j) + \varepsilon^j); (\hat{\mu}_{i_*,t} + \varepsilon_{i_*,t})\right\} \geq \mu_*(\bar{\theta}^j)$$

and

$$B_t^j(i; \theta_t^j) \leq \hat{\mu}_{i,t} + \varepsilon_{i,t} \leq \mu_i(\bar{\theta}^j) + 2\varepsilon_{i,t}.$$

Bringing the two conditions together we have

$$\mu_i(\bar{\theta}^j) + 2\varepsilon_{i,t} \geq \mu_*(\bar{\theta}^j) \Rightarrow 2\varepsilon_{i,t} \geq \Delta_i(\bar{\theta}^j),$$

which coincides with the (high-probability) bound on the number of pulls for $i$ using a *UCB* algorithm and leads to the statement by definition of $\varepsilon_{i,t}$. $\qquad\square$

*Proof of Lem. 12.* According to Lem. 10, a model $\theta$ can only propose arms in $\mathcal{A}_+^j(\theta; \bar{\theta}^j)$. Similar to the analysis of *mUCB*, $\theta$ is discarded from $\Theta_t^j$ with high probability after $t$ steps and $j$ episodes if

$$2(\varepsilon_{i,t} + \varepsilon^j) \leq \Gamma_i(\theta, \bar{\theta}^j).$$

At round $j$, if $\varepsilon^j \geq \Gamma_i(\theta, \bar{\theta}^j)/2$ then the algorithm will never be able to pull $i$ enough to discard $\theta$ (i.e., the uncertainty on $\theta$ is too large), but since $i \in \mathcal{A}_*^j(\theta; \bar{\theta}^j)$, this corresponds to the case when $\theta \in \widetilde{\Theta}^j(\bar{\theta}^j)$. Thus, the condition on the number of pulls to $i$ is derived from the inequality

$$\varepsilon_{i,t} \leq \Gamma_i(\theta, \bar{\theta}^j)/2 - \varepsilon^j.$$

$\qquad\square$

# E    Related Work

As discussed in the introduction, transfer in online learning has been rarely studied. In this section we review possible alternatives and a series of settings which are related to the problem we consider in this paper.

**Models estimation.** Although in *tUCB* we use *RTP* for the estimation of the model means, a wide number of other algorithms could be used, in particular those based on the method of moments (MoM). Recently a great deal of progress has been made regarding the problem of parameter estimation in LVM based on the method of moments approach (MoM) (Anandkumar et al., 2012c,a,b). The main idea of *MoM* is to match the empirical moments of the data with the model parameters that give rise to nearly the same corresponding population quantities. In general, matching the model parameters to the observed moments may require solving systems of high-order polynomial equations which is often computationally prohibitive. However, for a rich class of LVMs, it is possible to efficiently estimate the parameters only based on the low-order moments (up to the third order) (Anandkumar et al., 2012c). Prior to *RTP* various scenarios for *MoM* are considered in the literature for different classes of LVMs using different linear algebra techniques to deal with the empirical moments Anandkumar et al. (2012c,a). The variant introduced in (Anandkumar et al., 2012c, Algorithm B) recovers the matrix of the means $\{\mu(\theta)\}$ up to a permutation in columns without any knowledge of $\rho$. Also, theoretical guarantees in the form of sample complexity bounds with polynomial dependency on the parameters of interest have been provided for this algorithm. The excess correlation analysis (ECA) (Alg. 5 in Anandkumar et al. (2012a)) generalizes the idea of the *MoM* to the case that $\rho$ is not fixed anymore but sampled from some Dirichlet distribution. The parameters of this Dirichlet distribution is not to be known by the learner.[6] In this case again we can apply a variant of *MoM* to recover the models.

**Online Multi-task.** In the online multi-task learning the task change at each step ($n = 1$) but at the end of each step both the true label (in the case of online binary classification) and the identity of the task are revealed. A number of works (Dekel et al., 2006; Saha et al., 2011; Cavallanti et al., 2010; Lugosi et al., 2009) focused on this setting and showed how the samples coming from different tasks can be used to perform multi-task learning and improve the worst-case performance of an online learning algorithm compared to using all the samples separately.

**Contextual Bandit.** In contextual bandit (e.g., see Agarwal et al., 2012; Langford and Zhang, 2007), at each step the learner observes a context $x_t$ and has to choose the arm which is best for the context. The contexts belong to an arbitrary (finite or continuous) space and are drawn from a stationary distribution. This scenario resembles our setting where tasks arrive in a sequence and are drawn from a $\rho$. The main difference is that in our setting the learner does not observe explicitly the context and it repeatedly interact with that context for $n$ steps. Furthermore, in general in contextual bandits some similarity between contexts is used, while here the models are completely independent.

**Non-stationary Bandit.** When the learning algorithm does not know when the actual change in the task happens, then the problem reduces to learning in a piece-wise stationary environment. Garivier and Moulines (2011) introduces a modified version of *UCB* using either a sliding window or discounting to *track* the changing distributions and they show, when optimally tuned w.r.t. the number of switches $R$, it achieves a (worst-case) expected regret of order $O(\sqrt{TR})$ over a total number of steps $T$ and $R$ switches. Notice that this could be also considered as a partial transfer algorithm. Even in the case when the switch is directly observed, if $T$ is too short to learn from scratch and to identify similarity with other previous tasks, one option is just to transfer the averages computed before the switch. This clearly introduces a transfer bias that could be smaller than the regret cumulated in the attempt of learning from scratch. This is not surprising since transfer is usually employed whenever the number of samples that can be collected from the task at hand is relatively small. If we applied this algorithm to our setting $T = nJ$ and $R = J$, the corresponding performance would be $O(J\sqrt{n})$, which matches the worst-case performance of *UCB* (and *tUCB* as well) on $J$ tasks. This result is not surprising since the advantage of knowing the switching points (every $n$ steps) could always be removed by carefully choosing the worst possible tasks. Nonetheless, whenever we are not facing a worst case, the non-stationary *UCB* would have a much worse performance than *tUCB*.

Figure 9: Complexity and per-episode regret of *tUCB* over tasks.

# F  Numerical Simulations

|  | Arm1 | Arm2 | Arm3 | Arm4 | Arm5 | Arm6 | Arm7 |
|---|---|---|---|---|---|---|---|
| $\theta_1$ | 0.9 | 0.75 | 0.45 | 0.55 | 0.58 | 0.61 | 0.65 |
| $\theta_2$ | 0.75 | 0.89 | 0.45 | 0.55 | 0.58 | 0.61 | 0.65 |
| $\theta_3$ | 0.2 | 0.23 | 0.45 | 0.35 | 0.3 | 0.18 | 0.25 |
| $\theta_4$ | 0.34 | 0.31 | 0.45 | 0.725 | 0.33 | 0.37 | 0.47 |
| $\theta_5$ | 0.6 | 0.5 | 0.45 | 0.35 | 0.95 | 0.9 | 0.8 |

Table 1: Models.

|  | *UCB* | *UCB+* | *mUCB* |
|---|---|---|---|
| $\theta_1$ | 22.31 | 14.87 | 2.33 |
| $\theta_2$ | 23.32 | 15.58 | 8.48 |
| $\theta_3$ | 33.91 | 25.21 | 2.08 |
| $\theta_4$ | 17.91 | 11.17 | 3.48 |
| $\theta_5$ | 35.41 | 8.76 | 0 |
| avg | 26.57 | 15.11 | 3.27 |

Table 2: Complexity of *UCB*, *UCB+*, and *mUCB*.

In Table 1 we report the actual values of the means of the arms of the models in $\Theta$, while in Table 2 we compare the complexity of *UCB*, *UCB+*, and *mUCB*, for all the different models and on average. Finally, the graphs in Fig. 9 are an extension up to $J = 10000$ of the performance of *tUCB* for $n = 5000$ reported in the main text.

## Footnotes

[6]We only need to know sum of the parameters of the Dirichlet distribution $\alpha_0$.