[Reviews · NeurIPS 2013]

Submitted by Assigned_Reviewer_6

This paper considers transfer learning in a multi-armed bandit setting. The
model considered has a sequence of episodes, and in each episode, the vector of
distributions (one for each arm) is drawn iid from a discrete distribution.
In this setting, it is possible to exploit history to learn what this discrete
distribution is, and to use this information to reduce regret in each episode.
An algorithm is proposed that does this, and cumulative regret bounds are shown
for this algorithm.

Overall, I think this is an interesting paper, which makes a nice contribution
to the bandit literature. I think it also opens a number of questions into the
best way to do transfer learning in a bandit setting, and I hope the community
continues to investigate this thread.


Detailed Comments:
1. Line 132 to the end of section 3: As UCB does not require the specification of
a finite set of models, while mUCB does, and the regret bound for mUCB gets
worse as m increases, it would be worth discussing the dependence on m more
explicitly in this section. When m is large, it seems that the regret bound
for mUCB is worse than that of UCB.

2. Here is a follow-up question for future work: if we relax the assumption that m
is finite, and take \Theta to be the set of all distributions, the regret bound
in the paper goes to infinity and epsilon_{i,t} goes to infinity, which would
seem to cause mUCB to not perform well, even though we know it is possible to
have a non-trivial regret bound in this case. Is there a different algorithm
that takes advantage of knowing the set of models, but which retains good
behavior and a good regret bound as m goes to infinity? e.g., it would be nice
if the algorithm would become identical to UCB if we take \Theta to be the set
of all distributions.

3. I was somewhat surprised that the paper uses the Robust Tensor Power Method
to estimate the second, and third moments of the model distributions. It would
be worth discussing in the paper why this is done, rather than simply using the
corresponding sample means, and confidence bounds based on these sample means.

4. The algorithm doesn't seem to take advantage of any heterogeneity in rho.
If J is large, and some tasks are more common than others, it would seem
possible to learn rho, and to reduce regret for those tasks that are more
common. Perhaps this would be something interesting to explore in future work.

5. The main drawback in my mind of this method is that it requires knowing the
value of m. In practice, this would not be known. The paper would benefit of
some discussion of this point, and perhaps of some numerical experiments to
investigate misspecification of m.


Comments regarding clarity:
In a few places, the paper is not as clear as it should be. Here are some
points on which clarity can be improved:

1. Is it true that in the initialization phase, we may sample the tasks
non-sequentially, so that we take one sample for each arm for each task? This
was not clear from the text, and was only clear from Figure 2.

2. The second paragraph of section 2 needs to be edited for clarity:
1a. Line 082: the paper hasn't defined rho(theta) yet. It seems to be the probability that an arm is described by model theta.
1b. Line 077 and Line 081: we are conditioning on \theta being equal to
\bar{\theta}, but then \theta rather than \bar{\theta} appears on the right
hand side. Is one of these a constant dummy variable, and one a random
variable? Later it seems that \bar{\theta} is the random variable, but this is
unclear. Maybe it is unnecessary to have two variables here.
1c. Line 079-080: the paper explicitly states what the entries of M_2 are, but
not what the entries of M_3 are. Also, on line 078, it is stated, "Given two
realizations X^1 and X^2", but then we also use realization X^3 in the
definition of M_3.
1d. Line 095: "The only information available to the learning is the number of
models m, number of episode J and number of steps n per task." The reader
might naturally ask the question, "Does the learner also get to know \Theta?",
since the mUCB algorithm does, and algorithm that don't need to know the
model are not discussed until section 4.

3. Is there an assumption that rewards are bounded in [0,1]? I didn't see this
assumption stated anywhere, but it seems to be used in the proof of Theorem 1.

4. In Figure 5, please make the font bigger.
Summary: I think this is an interesting paper, which makes a nice contribution to the bandit literature. It addresses an interesting new question (transfer learning in the bandit setting) in an interesting way.

Submitted by Assigned_Reviewer_8

Summary: This paper studies online transfer learning in multi-armed bandits (MAB), where each instance of MAB encountered belongs to a finite model class. Instead of running single-task MAB algorithm, the paper shows how one can do better through sequential transfer across tasks. The paper first studies a variant of UCB1 (called mUCB): it assumes a set of MAB models is given as input, and improves on UCB1 by restricting the upper confidence bound calculation to the given models. Then, this algorithm is extended to the sequential transfer case (called tUCB), where the set of models is unknown, but has to be discovered by the algorithm. Here, the authors build on recent advances in tensor-based techniques to estimate the set of models. Under certain assumptions, tUCB enjoys lower per-task regret bounds than standard UCB1. Numerical experiments in synthetic problems verify the theoretical findings.

* Quality: The problem studied by the paper (online transfer learning in MAB) is new, and the proposed methods are very reasonable. While mUCB is a relatively straightforward variant, its extension to tUCB, where the set of models is unknown, is definitely nontrivial. The theory appears correct.

* Clarify: Overall the paper is written clearly, but the notation is very heavy and I wonder if it can be simplified. Also, there are a few places where the presentation can be improved (such as undefined quantities). Please see the detailed comments below.

* Originality: The problem formulation is new, to the best of my knowledge. The first new algorithm, mUCB, is a simple variant of UCB1, as mentioned before. But tUCB is sufficiently novel from previous bandit algorithms. While the analysis techniques are mostly based on existing literature (including MAB and the recent tensor-related work), its use of method-of-moment methods in the context of MAB is interesting and first of its kind.

* Significant: While the problem is interesting and important, the algorithm/analysis appear somewhat preliminary, for two reasons. One is the pretty strong dependence on the condition number of the second moment matrix M_2. The other is the assumption that the vectors {\mu(\theta)}_\theta are linearly independent. Please see the detailed comments below.

Detailed comments:

* The finite-sample convergence rate in Thm. 2 may be too loose in practice --- the strong dependence on #arms, #models, and the reciprocal of the smallest eigenvalue can make the bound too loose to be useful. One may wonder if these dependences are necessary, or are due to the non-tight analysis in this work.

* A more critical problem that limits the significance of the results is assumption 1, which requires linear independence of {\mu(\theta)}_\theta. Note that this is a set of m vectors of dimension K. To satisfy this assumption, it must be the case that m (number of all possible models) is no larger than K (number of arms). This restriction is both too strong and (I think) unnecessary.

* The current algorithms (mUCB and tUCB) are both variants of UCB1, with changes in the calculation of the confidence intervals. As acknowledged in the paper, a much better approach might be to consider the amount of information obtained by pulling an arm --- a suboptimal arm may result in larger regret in the short term but may help identify the current model more quickly. It is possible that, by considering such arms the restrictive assumption 1 is not needed.

* Title and in the paper: 'multi-arm bandit' should be 'multi-armed bandit'

* The sentence in line 62 (starting with 'This') does not seem to parse.

* Line 76: should X be realization of *arms*, or *reward* of the arms?

* Section 2: it seems more nature to swap paragraphs 2 and 3.

* Line 96: apparently, K is also part of the input to the learner.

* Line 117: the notation \Theta_+ should probably be \Theta_+(\bar{\theta}) to emphasize its dependence on \bar{\theta}.

* Line 189: 3 samples are not enough to *accurately* estimate the moments, but are enough to make the estimates *well-defined*.

* Line 211: 'finite sample complexity bounds' -> 'sample complexity bounds'? Sample complexity bounds are by definition about finite samples.

* Line 222: \sigma_max = max ..., instead of min...?

* Line 222: \lambda(\theta) seems undefined?

* Line 224 (assumption 1): why does one need \rho(\theta)>0? If \rho(\theta)=0 for some \theta, then the model is not going to affect the algorithm anyway.

* Line 269: what is alpha_0?

* Line 279: since \Gamma is defined as the absolute difference, its estimate should probably be: \hat{\Gamma} = max(0, current_definition_in_the_paper).

* Line 342: again, should 'arms' be 'rewards', in the last sentence of Thm. 4?

* Line 419: 'rises' -> 'raises'.
Summary: This is a first attempt on an intereting problem, but the contributions seem a bit preliminary at the current stage.

Submitted by Assigned_Reviewer_9

This paper introduces a model of transfer learning for bandit problems. There are m different unknown instances of a K-armed stochastic (i.i.d.) bandit problem. Time is divided in J episodes of n time steps each. At the beginning of each new episode, a new bandit instance is drawn from a fixed prior. Since J >> m, each instance occurs in several episodes, and the bandit algorithm has a chance of reusing knowledge from previous episodes. The paper uses a newly developed variant of the method-of-moments to achieve this transfer learning with guarantees that the resulting regret bound never worse than UCB's regret bound and approaches that of the oracolar algorithm which knows the set of instances.

This is a solid paper applying new estimation techniques to solve the bandit problem in a new and complex setting. The theoretical results are strong, and the experiments nicely support them.

My main criticism is the following: the tensor-based estimation techniques are relatively new. The paper is quite dry in this respect, and does not do a great job in providing an accessible explanation of such techniques. The reader is referred over and over to results published elsewhere.
The bounds themselves are not easy to interpret, although in this case the remarks give good intuitions.

Assumption 1 limits the scope of application of the results to cases where m < K. Any hope of lifting this assumption with the current techniques? How strongly do the results rely on the assumption that each task in an episode is an independent draw from the prior?

Line 74: (\theta) missing in the definition of \Delta_i(\theta) ?

Line 76: reward distributions have a bounded support, I believe

Line 81-82: prior \rho is used before definition

Line 86-87: What is n in the upper limit of the sum?

====================================================

I have read the authors' rebuttal.
Summary: A solid set of results. Some assumptions are rather limiting. The presentation could be made more friendly to the reader not familiar with the method of moments.
Author Feedback

Author rebuttal: We thank the reviewers for the very helpful feedback. Due to lack of space we will focus on the most central questions and comments but we are happy to address all small suggestions in a camera ready version. We also will correct the typos and improve the clarity in the final version.

Rev. 6
1,2. A more precise definition of eps_{i,t} contains a log(min(m,K)n^2/delta)) term. This implies that the logarithmic term in eq.3 only depends on min(m,K) and doesn't keep growing as m tends to infinity. Furthermore, the set of actions A+ is never larger than K. Thus even when \Theta is the set of all distributions, mUCB is guaranteed to perform as well as UCB. We preferred to avoid reporting min(m,K) in the bound for the sake of readability but please refer to proof of Lem.2 in the supplementary material to see that all the results still hold unchanged.

3. Unfortunately using the sample means wouldn't provide enough accurate estimates of the bandit problems for two reasons. (i) Since we don't know the identity of the current model, at the end of the episode, we wouldn't know which model to update with the current estimates. (ii) When n (number of steps per episodes) is small, the sample means in the current episode are very inaccurate, whereas RTP can be used to estimate the set of MAB model parameters even if each episode is very short: only three samples per-arm (at each episode) is sufficient for RTP to converge. We will make these points clearer in the paper.

4. We agree with the reviewer. Nonetheless, the RTP method implicitly models \rho as well although we do not use it explicitly in the algorithm. Probably an explicit use of \rho might be more beneficial in a Bayesian regret setting.

5. We agree that in some problems m is not known to the learner and needs to be estimated. This can be done by estimating the rank of matrix M_2 which is equal to m (See Newey and Smith (2004) for matrix rank estimation). We will include a few lines in the final version to clarify this issue and suggest it as a future work.


Rev.8
"The finite-sample convergence rate in Thm. 2 may be too loose in practice"

We agree that these bounds may be looser than desirable for practical applications. However, we believe that all these dependencies is necessary though the order may be improved. Our bounds are very close to the-state-of-the-art for the method of moments in terms of dependency on the above-mentioned parameters (see discussion in the Remark after Theorem 2). It is an interesting issue for future work as to whether the bounds can be tightened given additional structure present in certain application domains.

"A more critical problem .... is linear independence of {\mu(\theta)}_\theta."
The assumption that {mu(theta)} are linearly independent is crucial for all variants of method of moments including RTP. While there are other methods, like EM, for learning latent variable models, such models often converge to local optima and have no finite sample bounds. To the best of our knowledge RTP is the most advanced method for learning latent variable models. It is an open question whether it is possible to remove this assumption, since linear dependency might prevent the method to "distinguish" the models and properly estimate their means. Although relaxing the assumption of linear independence is out of the scope of this paper (our primary focus is to learn latent variable models to enable transfer of knowledge in MABs), in the experiments we considered a challenging scenario where the smallest singular value is relatively small and still achieved good results.

"To satisfy this assumption (linear independency), it must be the case that number of all possible models is no larger than number of arms."
we think, even with this assumption, our method still covers many problems in which sequential transfer of knowledge can make a significant difference. This is due to the fact that the transfer of knowledge, in our setting, is most beneficial when the number of models is small in compare to the number of actions since the regret of mUCB scales with the number of models than the number of actions: mUCB has a dependence on the number of optimal arms K and models m which is at most min(m,K), compared to UCB which is O(K).

"The current algorithms (mUCB and tUCB) are both variants of UCB1…"
As we discussed in the paper, we certainly agree considering the information obtained by pulling an arm could be a promising alternate approach. Nonetheless, this could translate in a better regret bound for mUCB but it would not impact the quality of the estimates returned by RTP. On the other hand, if we decide to only rely on the sample means instead of using RTP, this might have some drawbacks as discussed in comment 3 of reviewer 6. We think this is an interesting direction for future work.

"why does one need rho(theta)>0?"
This assumption is needed for the convergence of RTP. Although if we estimate the effective number of models by estimating the rank of matrix M_2, we can remove this assumption (See reply to rev.6/comment 5).

Rev.9
"the tensor-based estimation techniques are relatively new"
We agree that we need a more detailed description of RTP in the final version of the paper. We will use part of the material in Sections C and E of the supplementary material.

"Assumption 1 limits the scope of application…"
Please refer to comments to rev.8

"How strongly do the results rely on the assumption that each task in an episode is an independent draw from the prior?"
We use the iid assumption for the sake of simplicity in the analysis but we believe that it would be possible to replace the iid assumption with some weaker assumptions, such as the assumption that tasks are drawn from a Markov chain. This may require using more sophisticated concentration tools for stationary processes (see e.g., Meir 2000).